

**Fault-controlled dolomitization in the Montagna dei Fiori Anticline (Central Apennines, Italy): Record of a dominantly pre-orogenic fluid migration**

Mahtab Mozafari[1], Rudy Swennen[2], Fabrizio Balsamo[1], Hamdy El Desouky[2,3], Fabrizio Storti[1] and Conxita Taberner[4]

1- NEXT - Natural and Experimental Tectonics Research Group - Department of Chemistry, Life Sciences and Environmental Sustainability, University of Parma, Italy.

2- Department of Earth and Environmental Sciences, KU Leuven, Belgium.

3- Geology Department, Faculty of Science, Menoufia University, Menoufia, Egypt

4- Shell Global Solutions International B.V., Amsterdam, The Netherlands

Correspondence: Mahtab Mozafari (mahtab_mozafari@yahoo.com) and Fabrizio Storti (Fabrizio.Storti@unipr.it)

**Abstract**

The Lower Jurassic platform and basinal deposits exposed in the Montagna dei Fiori Anticline (Central Apennines, Italy) are pervasively affected by dolomitization. Based on the integration of field work, petrography, and geochemistry, two fault-related dolomitization events were recognized and interpreted as occurred before and during the Apenninic orogeny, respectively. Fluid inclusion analysis indicates moderate to elevated salinity values of 3.5 to 20.5 and 12.8 to 18.6 eq. wt. % NaCl, in the first and the second event, respectively. The estimated salinities, in combination with $\delta^{18}O$ values and $^{87}Sr/^{86}Sr$ ratios, suggest significant involvement of evaporitic fluids in both events, most likely derived from the underlying Upper Triassic Burano Formation. In addition, the $^{87}Sr/^{86}Sr$ ratios up to 0.70963 suggest the circulation of deep-sourced fluids that interacted with siliciclastics and/or the crystalline basement during the dolomitization events. The first dolomitization event which is also considered as the most pervasive one started prior to the significant burial conditions, as reflected in homogenization temperatures of their fluid inclusions being mostly below about 40-50°C. Two major dolomite types (D1 and D2) were recognized as pertaining to this event, both postdated by high amplitude bed-parallel stylolites. This relationship supports a syn-burial, pre layer-parallel shortening dolomitization, interpreted as controlled by the extensional fault pattern affecting the carbonate succession before its involvement in the Apenninic thrust wedge. A possible geodynamic



framework for this dolomitization event is Early to Late Jurassic rift-related extensional
tectonism.
The second dolomitization event initiated with a dolomite type (D3) characterized by a slight
temperature upturn (up to 73°C), followed by a second type (D4) with markedly higher
homogenization temperatures (up to 105°C), interpreted as associated with the inflow of
hydrothermal fluids, possibly related to major changes in the permeability architecture of faults
during early- to syn-thrusting and folding activity. Eventually, D4 was overprinted by a late
generation of dolomite veins (D5) interpreted as associated with late orogenic extensional
faulting in the backlimb of the Montagna dei Fiori Anticline. Based on the timing of deformation
in the Montagna dei Fiori Anticline, D3 to D5 dolomitization likely occurred in Late Miocene to
Pliocene times. The findings regarding characteristics and timing of dolomitization here
illustrates the long-term controlling role of the eveporitic detachments in dolomitization process.
Our data shows the Mg-rich fluids most likely derived from these evaporites may prime the
tectonically involved successions for repeated dolomitization, and formation of potential
reservoirs in sequential tectonic modifications (extensional vs. compressional).
**1 Introduction**
Fault-controlled dolomitization has been the focus of attention in many studies during the last
decades due to its influential role in modifying the petrophysical properties of rocks and, hence,
anisotropy in fluid migration pathways, and, ultimately on reservoir quality (e.g. Purser et al.,
1994; Montanez, 1994; Zempolich and Hardie, 1997; Vandeginste et al., 2005; Davies and
Smith, 2006; Sharp et al. 2010). The mechanical and hydrological behaviour of fault zones are in
turn influenced by fluid-rock interactions and diagenetic modifications (e.g. Gale et al., 2004;
Laubach et al., 2010; Clemenzi et al., 2015). It follows that the mutual interplay between fault
activity and fluid-driven rock-fluid interaction can trigger dolomitization of carbonates and,
consequently, variations in physico-chemical properties of fluids through time and space.
Leaking or sealing behaviours of fault zones during deformation are key controls for fault-related
fluid circulation. A detailed understanding of such an interplay is thus necessary to improve our
capability of making reliable predictions of fault-related dolomitization in carbonate reservoirs.
Studying outcrop analogues provides fundamental support to meet this requirement (e.g.
Swennen et al., 2012; Dewit et al., 2014; Bistacchi et al., 2015).





The Lower Jurassic to Lower Cretaceous Umbria-Marche passive margin carbonate
succession, in the Central Apennines (Italy), is intensely affected by localized dolomitization
both in the onshore fold-and-thrust belt and in offshore foredeep and foreland areas (e.g. Murgia
et al., 2004; Pierantoni et al., 2013). The dolomitized intervals are well-exposed in the core of the
Montagna dei Fiori Anticline (e.g. Ronchi et al., 2003), where the dolomitized Lower Jurassic
intervals (Calcare Massiccio, Bugarone and Corniola Formations) and their relationships with
fault zones allow to study the mutual influence between deformation structures and dolomitized
intervals (Fig. 1). These intervals, known as the Castel Manfrino Dolostones (Crescenti, 1969;
Mattei, 1987; Koopman, 1983), have been previously studied by Ronchi et al. (2003) at its
reference section, the Castel Manfrino location, in the central sector of the Montagna dei Fiori
Anticline (Fig. 2). A fault-controlled dolomitization model and the relative timing of
dolomitization were proposed by Ronchi (2003). Recent re-evaluation of dolostone distribution
in the Montagna dei Fiori Anticline (Storti et al., 2017a), showed that the dimension of the
dolomitized geobodies (Fig. 2) is much more significant than what was previously mapped by
Mattei (1987). Dolostones are distributed within fault damage zones and in the laterally adjacent
carbonate rocks, and in intersection areas between fault sets, for a total area in map view of more
than 1.5 km$^2$ (Storti et al., 2017a).
The structural pattern of the Montagna dei Fiori Anticline documents the overprinting of
extensional and contractional deformation along major fault zones. Although challenging, the
preserved structural framework in this anticline provides an opportunity to study the direct but
complex regional tectonic controls on dolomitization in carbonate successions undergoing
multiple deformation events, from rifting to folding and thrusting. This contribution integrates
field mapping, new petrographic, geochemical, and microthermometric analyses, with structural
studies to characterize the temporal record of fault-controlled diagenetic phases and, more
specifically, dolomitization in the carbonatic succession outcropping in the Montagna dei Fiori
Anticline. Therefore provides insights into the structural controls on regional fluid flow and their
chemical evolution through time. These findings might be of relevance for exploration and
reservoir quality prediction onshore and offshore the Apennines and Southern Alps. Moreover,
this work provides additional evidence of the potential influence of fluids derived from
evaporitic detachment levels in modifications of geochemical trends and petrophysical properties
of the overlying carbonate rocks.





## 2 Geological setting


The Montagna dei Fiori Anticline is a NNW-SSE striking, thrust-related fold located at
the mountain front of the Central Apennines (Fig. 1). The geodynamic evolution of the
Apennines is generally known to be the result of the superposition of NE-SW compression (in
present-day geographic coordinates), related to the convergence between Eurasia and Africa
plates since Late Cretaceous times (Elter et al., 1975; Dewey et al., 1989; Patacca et al., 1992),
on a rifting-related tectono-sedimentary architecture produced by Early Jurassic extension (e.g.,
Centamore et al., 1971). In such a framework, the Central Apennines developed during Miocene
to Plio-Pleistocene times (e.g. Parotto and Praturlon, 1975; Barchi et al., 1998; Mazzoli et al.,
2002; Bollati et al., 2012).
The Central Apennines involves the Umbria-Marche succession, which essentially
includes Triassic to Miocene carbonates and marls, covered by Miocene to Pliocene syn-
orogenic clastic sediments (Fig. 1). The pre-orogenic succession, from bottom to top, includes
Late Triassic evaporites, dolomites and limestones (Burano Formation), Early to Late Jurassic
platform and basinal limestones and dolostones (Calcare Massiccio, Corniola, Rosso
Ammonitico, Calcari a Posidonia and Calcari ad Aptici Formations), and Cretaceous to Early
Miocene basinal carbonates (Maiolica, Marne a Fucoidi, Scaglia and Biscaro Formations). In
general, the lower part of Burano Formation is overlaid by the fluvio-deltaic siliciclastics of the
Verrucano Formation (Middle-Late Triassic) (Tongiorgi et al., 1977; Ghisetti and Vezzani, 2000;
Tavani et al., 2008). Nevertheless, the existence of these siliciclastics in the Montagna dei Fiori
area is not yet proved. Syn-orogenic deposits include Miocene marls and turbiditic sandstones
(Marne con Cerrogna and Laga Formations) (Artoni, 2013 and references therein).
The deposition of the Calcare Massiccio Formation, dated as Hettangian-Sinemurian and
with a total thickness varying between 300 to 700 m (Pialli, 1971), records an important
extension pulse in the evolution of Tethyan rifting. The following facies are observed in the
lower part of the Calcare Massiccio Formation: oncoid-rich peloidal pack- to grainstones in
alternation with peloidal wacke- to packstones including horizons of algal bindstones (Calcare
Massiccio A; Brandano et al., 2016). The upper part is made up of beds of skeletal and coated
grain wacke- to grainstones including microoncoids, echinoderms, calcareous and siliceous
sponges, bivalves, gastropods and ammonites (Calcare Massiccio B; Brandano et al., 2016). The
lower part has been interpreted as having been deposited in a peritidal environment, while the



upper part corresponds to lower to middle shelf depositional environments, characterized by a
general deepening upward trend associated with extensional faulting and drowning of the
platform, coupled with subsidence and deposition of the overlying Corniola Formation in the
pelagic areas (Sinemurian-Toarcian; Colacicchi et al., 1975; Morettini et al., 2002; Bosence et
al., 2009; Marino and Santantonio, 2010; Brandano et al., 2016). Overall, the Early Jurassic
rifting led to the growth of the Calcare Massiccio Formation in a carbonate platform setting,
followed by faulting and drowning, and development of pelagic intrabasins filled by syn-rift
sediments (Fig. 1c; Bernoulli et al. 1979; Santantonio and Carminati, 2011). Condensed pelagic
limestones of the Bugarone Formation (Lower Pliensbachian-Lower Tithonian; Bugarone Group
in Pierantoni et al., 2013) occur at the top of the Calcare Massiccio Formation where it formed
fault-controlled highs marking the regional drowning of the carbonate platform (Santantonio and
Carminati, 2011). In the Montagna dei Fiori, the geologic framework of the outcropping Calcare
Massiccio Formation is still a matter of debate between a fault-related tectonosedimentary
pattern (Mattei, 1987; Storti et al., 2017b), and a gravity-driven, olistolith hypothesis (Di
Francesco et al, 2010; Santantonio et al., 2017). However, recent detailed work in the Salinello
valley (Storti et al., 2017a; 2018) documented that major outcrops of Calcare Massiccio are
bounded by mostly ~ E-W and ~ N-S striking fault zones showing extensional kinematics and
dominantly affecting the Jurassic rocks older than the Maiolica Formation (Fig. 2A, e.g. sites 1
to 4). Overprinting relations indicate that ~ E-W deformation structures are systematically
younger than the ~ N-S ones. Similar trends were observed in syn-rift fault zones in other
anticlines of the Central Apennines (e.g. Cooper and Burbi, 1986; Alvarez, 1989; Chilovi et al.,
2002).

Such a tectonosedimentary inheritance was involved in the growth of the Montagna dei
Fiori Anticline, which initiated during the Late Miocene (Mazzoli et al., 2002; Artoni, 2003) and
progressively evolved into the upper thrust sheet of a well-developed antiformal stack until Plio-
Pleistocene times (e.g. Ghisetti et al., 1993; Calamita et al., 1994; Artoni, 2013). A major
structural feature trending parallel to the Montagna dei Fiori Anticline and dissecting it is the
Montagna dei Fiori Fault, a NNW-SSE striking extensional fault system cutting at high angle
through the folded footwall rocks, typically at the forelimb-crest transition (Figs. 1, 2). This fault
system juxtaposes intensely deformed Late Miocene sediments in the hanging wall, against
dolomitized and undolomitized Lower Jurassic and Cretaceous limestones in the footwall (Figs.



1 and 2). The development of the Montagna dei Fiori Fault has been alternatively interpreted as
either a pre- (e.g. Calamita et al., 1994, Mazzoli, 2002; Scisciani et al., 2002) or late-folding
(Ghisetti and Vezzani, 2000) feature. More recently, the origin of the Montagna dei Fiori Fault
has been ascribed to the mutual interaction between horizontal shortening and uplift, and
episodic gravitational re-equilibration during antiformal stacking underneath the anticline during
Plio-Pleistocene times (Storti et al., 2018).

**3 Methodology**

The stratigraphic and deformational features of dolostones were analyzed in more than 60
outcrops. The distribution of dolomitized intervals as well as their cross-cutting relationships
with bedding planes, stylolites, veins and structures were ground-truthed and sampled. For
petrographic analyses, 130 polished thin sections were studied with standard petrographic
methods (transmitted and UV-fluorescent light microscopy). Dolomite crystal morphology and
texture is based on the classification proposed by Sibley and Gregg (1987).
The rock slabs and thin sections were stained using Alizarine Red S and potassium
ferricyanide (Dickson, 1966) to discriminate dolomite from calcite and evaluate their iron
content. Cold cathodoluminescence microscopy (CL) was carried out on representative thin
sections (n = 80) at KU Leuven University (Belgium) using a Technosyn cathodoluminescence
device (8-15 kV, 200-400 μA gun current, 0.05 Torr vacuum and 5 mm beam width).
$\delta^{13}C$ and $\delta^{18}O$ analysis were carried out on 117 samples. Powder samples (150 - 200 μg)
were obtained by applying a New Wave Research micromilling device and a dental drill at KU
Leuven University (Belgium). The analysis was conducted at Parma University (Italy) and the
Friedrich-Alexander-Universität (Erlangen-Nürnberg, Germany) laboratories using Finnigan
DeltaPlus V and ThermoFinnigan 252 mass spectrometers, respectively. The carbonate powders
were reacted with 100% phosphoric acid at constant temperature of 75°C. Several additional $CO_2$
reference gases (NBS18, NBS19, MAB99, and a pure Carrara marble) with known isotopic ratio
were analyzed during the measurements to determine the $\delta^{13}C$ and $\delta^{18}O$ values of the sample.
Reproducibility was checked by replicate analysis of laboratory standards and was better than
±0.1‰ for $\delta^{13}C$ and ±0.2‰ for $\delta^{18}O$ at Parma University and ±0.04 for $\delta^{13}C$ and ±0.05‰ for
$\delta^{18}O$ at Friedrich-Alexander-Universität. Oxygen isotope composition of dolomites was
corrected using the acid fractionation factors given by Rosenbaum and Sheppard (1986).
Duplicate homogeneous samples measured in both labs for inter-laboratory reproducibility show



$\delta^{13}C$ and $\delta^{18}O$ values within the acceptable range of error deviation (±0.1‰) both for $\delta^{13}C$ and
$\delta^{18}O$. All carbon and oxygen values are reported in per mil, relative to the "Vienna PDB scale"
(V-PDB).

A total number of 21 samples were analyzed for their $^{87}Sr/^{86}Sr$ ratios. The analyses were

conducted at the Department of Analytical Chemistry, Ghent University (Belgium) and at the
Vrije Universiteit Amsterdam (the Netherlands). NIST SRM 987 was used as the international Sr
standard in both labs. At Ghent University, 15 sample powders (20 mg) were collected using a
dental drill device. The $^{87}Sr/^{86}Sr$ ratio measurements were performed using a Thermo Scientific
Neptune Multi-collector Inductively Coupled Plasma Mass Spectrometer (MC-ICP-MS)
instrument. Within the external precision, repeated analyses of the international Sr standard
yielded an average $^{87}Sr/^{86}Sr$ ratio of 0.710271 ± 0.000023 (2SD, n = 43), in agreement with the
accepted $^{87}Sr/^{86}Sr$ ratio of 0.710248 for this reference sample (Thirlwall, 1991). At Vrije
Universiteit Amsterdam, 6 sample powders (2 - 3 mg) were collected using a New Wave
Research micromilling device. Analyses were performed using a ThermoElectron Triton plus
TIMS instrument. In order to monitor and document the system's performance, repeated analyses
of the international Sr standard (n = 58) were carried out on load sizes of 10 ng and 100 ng which
yielded average $^{87}Sr/^{86}Sr$ ratios of 0.710245±0.000022 (2SD) and 0.710242±0.000008 (2SD),
respectively. In both labs mass discrimination correction was performed via internal
normalization using Russell's exponential law and the accepted value (0.1194; Steiger and Jager,
1977) of the invariant $^{86}Sr/^{88}Sr$ ratio.

Fluid inclusion microthermometry analysis was performed on 11 doubly polished wafers

(80-130 μm in thickness). Measurements were carried out at Parma University (Italy) using
Linkam THMSG-600 and Linkam MDS-600 heating-cooling stages coupled with a Leica DM
2500 microscope. The stages were calibrated by synthetic Syn Flinc$^{TM}$ fluid inclusion standards.
A 100x objective was used during the microthermometry runs of the small inclusions. The
microthermometry data were collected following the Fluid Inclusion Assemblage (FIA) approach
described in Goldstein and Reynolds (1994) for carbonate minerals. The salinities are reported in
equivalent weight percent NaCl (eq. wt. % NaCl) and were calculated based on the equation of
Bodnar (1993).

In order to perform a high resolution petrography, Scanning Electron Microscope (SEM)

and Back-scattered Scanning Electron Microscope (BSEM) analyses were conducted using a



Jeol 6400 Scanning Electron Microscope (SEM) equipped with an Oxford EDS (Energy
Dispersive System). Operating conditions were 15 kV and 1.2 nA, electron beam about 1 μm in
diameter and 100 s counting time; errors are ±2-5% for major elements and ±5-10% for minor
components. The analysis focused mainly on detecting possible dolomite crystals inside the bed
perpendicular stylolites affecting the Cretaceous Scaglia Formation.
**4 Results**
**4.1 Field observation and distribution of the dolomitized bodies**
Dolomitization affected the Calcare Massiccio, Bugarone and Corniola Formations.
There is no evidences of dolomitization in the overlying and immediate surrounding successions
(e.g. Maiolica and Scaglia Formations), though the base of Maiolica Formation is reported as
dolomitized in the Central Apennines onshore (e.g. Pierantoni et al., 2013) and offshore areas
(Murgia et al., 2004). Dolomitized intervals are folded in the forelimb of the Montagna dei Fiori
Anticline and are abruptly truncated by the Montagna dei Fiori Fault, which juxtaposes them
against intensely foliated Scaglia, Bisciaro and Marne con Cerrogna Formations (Figs. 2 and 3).
The distribution of dolomitized intervals is wider in the Salinello valley (Figs. 1B, 2A). In the
Corano Quarry location, dolomitization occur in the Calcare Massiccio and Bugarone
Formations only as meter-sized dolostone geobodies in the footwall of the Montagna dei Fiori
Fault (Fig. 4).
Dolostone breccias in fault cores is typically clast-supported, with angular and
millimeter- to centimeter-sized fragments (Fig. 3C), changing to crackle breccia (Woodcock and
Mort, 2008) away from the master slip surface. In the proximity of the master slip surface,
dolostone fragments are sporadically overprinted by millimeter-sized dolomite veins. The
breccia fragments, where cemented, are commonly surrounded by calcite.
Dolomitization does not follow a systematic pattern. In some outcrops, dolomitization
fronts show irregular outlines following, but also cross-cutting, the bedding surfaces (Fig. 5).
Dolomitized intervals vary in thickness from few meters to hundred meters affecting the totality
of the exposed Calcare Massiccio and only the lower part of Corniola Formation, where no clay
interlayers are present. Dolomitized intervals in the Corniola Formation have a darker color
relative to the host rock and are systematically more fractured than the hosting limestone. High
amplitude (> 1 mm) bed-parallel stylolites are clearly visible in both limestones and dolostones



(Fig. 5). However, in some dolostones only ghosts of stylolite traces can be seen. The dolostones
locally contain porosity, appearing as millimetre- to centimetre-sized pores.
**4.2 Petrography**
**4.2.1 Early calcite cementation**
The early diagenetic products in the studied intervals are generally non-ferroan calcite
cements. The first calcite cements precipitated following a phase of bioclast micritization (*sensu*
Bathurst, 1975) in grain supported intervals. In chronological order, they include: 1) fibrous
cements (FC) riming the bioclasts, mostly in the peloidal facies of the Calcare Massiccio
Formation (Fig. 6A). These cements are dull to non-luminescent under cathodoluminescence; 2)
mosaic cements (MC), commonly fill the intergranular pore spaces (Fig. 6B), and also occur as
syntaxial overgrowths on echinoderm fragments. These cements exhibit deformation twining and
show well-developed dull and orange concentrically-zoned cathodoluminescence pattern (Figs.
6C and D). They contain only mono-phase all-liquid inclusions. All of these cements are
postdated by high amplitude bed-parallel stylolites.
**4.2.2 Dolomitization**
All the dolomite types are non-ferroan and dominantly fabric destructive. The two first
dolomite types (D1 and D2) are the dominant dolomite types in the studied outcrops. These
dolomites are distributed within the damage zones of the ~ N-S and E-W Jurassic rift-related
extensional faults and, in places, displaced by them (Fig. 2A, site 1). The third and fourth
dolomite types (D3 and D4) are mainly observed within the damage zone of the Montagna dei
Fiori Fault (NNW-SSE), and appear only as dolomitic pockets overprinting D1 and D2 at the
proximity of the ~ N-S and E-W extensional faults. The fifth dolomite type (D5) is found only
within the brecciated zones associated with the Montagna dei Fiori Fault damage zone. The
distinctive petrographic features of the recognized dolomite types are summarized below:
**Dolomite 1 (D1)** is a replacive dolomite which commonly appears as dispersed rhombs and
aggregates, and locally rims fracture walls cemented by calcite (CV1) (Figs. 6E and F). D1
postdates the micritic envelopes and early calcite cements, and predates high amplitude bed-
parallel stylolites (Figs. 6G and H). The crystals are fine to medium sized (< 350 μm) and
consists of relatively turbid, solid-inclusion rich, well-developed euhedral to subhedral crystals,
with red luminescence, occasionally developing a concentrical zonation.



**Dolomite 2 (D2)** is a replacive dolomite (Figs. 7A and B), infrequently occluding existing pore spaces. Like D1, it also frequently predates high amplitude bed-parallel stylolites (Figs. 6G and H. D2 generally exhibits a tightly packed texture with no or little intercrystalline porosity. The crystals are medium to coarse sized (≤ 500 μm) including a turbid core followed by a transparent subhedral to anhedral rim and trace quantities of saddle dolomite developing swiping extinction. In some crystals one additional turbid zone rich in solid and fluid inclusions is present. Cathodoluminescence observations enabled to recognize the presence of D1 in their turbid cores. D2 crystals are characterized by zones of bright red-pink luminescence separated by purple luminescence zones.

**Dolomite 3 (D3)** is present as small localized bodies in the Calcare Massiccio (at the Castel Manfrino reference section), in the Corniola Formation (at the Osso Caprino Road), and in the Calcare Massiccio and Bugarone Formations (at the Corano Quarry). In the Corano Quarry the dolomitized Bugarone and Calcare Massiccio Formations are in the footwall of the Montagna dei Fiori Fault; and juxtaposed to the undolomitezed, intensely foliated Scaglia Formation (the hanging wall). Within the Bugarone Formation in this fault damage zone, D3 locally cements the millimeter-sized angular breccias that are in turn affected by fault parallel stylolites (Figs. 7C and D). The SEM and BSEM analysis performed on the samples from the immediate adjacent Scaglia Formation within the aforementioned fault damage zone did not indicate the presence of any dolomite in this formation. D3 crystals are mostly transparent euhedral to anhedral (< 300 μm), with minor development of saddle morphologies in larger crystals (> 500 μm) (Figs. 7E to H). The euhedral to anhedral crystals are generally replacive, displaying a faint core, which compared to previous dolomite types has fewer solid inclusions. The saddle crystals are occasionally replacive but majorly appear as cement in fractures. They display typical curved and slightly serrated crystal terminations with swiping extinction. These saddle dolomites were only observed in the Castel Manfrino reference section. D3 generally exhibit a dull purple color with bright orange zones and subzones in core and/or rims when viewed under cathodoluminescence (Figs. 7E to H).

**Dolomite 4 (D4)** appears as a matrix replacive and dolomite cement surrounding porosity and locally recrystallizing D1 and D2 (Figs. 8A to F). D4 also occludes bed parallel shear fractures and appears along the bed parallel stylolites (Figs. 9A to D). In the Castel Manfrino reference section, some intercrystalline vuggy porosity is filled with fine dolomite rhombs including D4



with relics of D2 within their core (Figs. 8E and F). The porosity may be preserved or partially to
completely filled by CV4. D4 crystals have a turbid, solid-inclusion rich core and transparent
rim. They are fine to medium sized (< 200-350 μm), presenting subhedral to infrequent euhedral
crystals. D4 exhibits a distinct luminescence pattern including a purple zone and an irregular
green subzone.
**Dolomite 5 (D5)** occurs as crystals cementing micro-veins that cross-cut precursor dolomite
types including dolomitic breccia fragments. In cemented breccias, D5 is postdated by CV3. D5
is transparent, anhedral and is characterized by a bright red luminescence (Figs. 9E and F).
**4.2.3 Late calcite cementation**
Four generations of calcite veins postdating dolomitization have been identified (Figs. 10
and 11): 1) Calcite vein 1 (CV1) occurs only in Calcare Massicio limestones and is represented
as centimeter-sized, strata-bound, bedding-perpendicular veins with irregular fracture walls,
exhibiting white color in the outcrops. They postdate the first dolomite type (D1) riming the
same fractures that abut the high amplitude bed parallel stylolites. CV1 often show blocky to
elongated crystal morphologies and displays well-developed deformation twinning planes (Type
II of Burkhard, 1993). This calcite exhibits concentrical zonation and dull zones alternate with
orange luminescence zones (Figs. 11A and B). 2) Calcite vein 2 (CV2) exclusively occurs in the
intensely deformed Scaglia Formation within the fault damage zones and correspond to tension
gashes associated with stylolites (*sensu* Nelson, 1981). CV2 veins are mostly recorded in foliated
shear deformation zones with well-defined S-C fabrics, exhibiting blocky, elongated to fibrous
shapes with strongly developed tightly spaced deformation twinning planes (Type II of
Burkhard, 1993). CV2 displays yellow to orange luminescence with locally darker sector zones.
The yellow to orange luminescence characteristic of CV2 is comparable with those of encasing
Scaglia host rocks (Figs. 11C and D). 3) Calcite vein 3 (CV3) occurs as cement, filling the
extensional faults master plane and isolated veins within the extesional fault damage zones. CV3
cements the brecciated fault-infillings containing angular fragments of host rock limestones,
dolostones and earlier calcites. In the brecciated zones at the backlimb of the anticline
(Montagna dei Fiori Fault), CV3 postdates the last dolomitization phase (D5) with no evidence
of physical disruption. CV3 exhibits a white to translucent color in hand specimen. The crystals
are blocky with no or weakly developed deformation twinning planes, and are characterized by a
dark orange to brown luminescence with distinct darker sector zones (Figs. 11E and F). 4)




Calcite vein 4 (CV4) exists as isolated veins, pore-filling as well as breccia cements postdating
all the preceding dolomites and calcites. The breccia fragments are more often dolostones. CV4
has a white to translucent white color in hand specimen with blocky crystal morphology and no
evidence of subsequent deformation (e.g. deformation twining planes), and is characterized by
distinct concentrical zonation (Figs. 11G and H).
**4.3 Geochemistry**
**4.3.1 Carbon and oxygen stable isotopes**
The carbon and oxygen stable isotopic data ($\delta^{13}$C and $\delta^{18}$O) of host rocks, dolomites and
calcites are given in Table 1 and shown in Figures 12A and B. The marine stable isotopic
compositions reported by Veizer et al. (1999) were used as marine reference values.
Accordingly, Lower Jurassic marine limestones are characterized by $\delta^{13}$C values of -0.5 to
+4.5‰ and $\delta^{18}$O values of -2.5 to +1.0‰ V-PDB. The $\delta^{18}$O values of the marine dolomites are
known to be 3-4‰ V-PDB more enriched than those of co-genetic marine limestones (Land,
1980; Major et al., 1992; Horita, 2014). Both $\delta^{13}$C and $\delta^{18}$O values of the host rocks are within
the expected range of the Lower Jurassic marine limestones but the Corniola host rocks show
slightly lower values comparing to those of Calcare Massiccio. In the Calcare Massiccio host
rocks, the $\delta^{13}$C values plot between +2.4 and +3.1‰ and $\delta^{18}$O values are within the range of -1.6
and 0.0‰ V-PDB. The $\delta^{13}$C values in the Corniola host rocks are +2.0 and +2.5‰ while the
$\delta^{18}$O values are -3.1 to -1.4‰ V-PDB. The $\delta^{13}$C and $\delta^{18}$O values of the Scaglia host rocks range
between +1.0 to +3.3‰ for $\delta^{13}$C and -2.2 to -1.0‰ V-PDB for $\delta^{18}$O. The obtained values are
characterized in the mean range of Upper Cretaceous to Paleogene marine limestones (Veizer et
al., 1999; +1.0 to +4.5‰ for $\delta^{13}$C and -4.0 to +2.0‰ V-PDB for $\delta^{18}$O).
The $\delta^{13}$C values of CV1 are between +1.6 and +2.1‰ which plot within the range of
reference values (Jurassic) but are slightly lower than the surrounding host rock values. The $\delta^{18}$O
values are between -4.7 and -2.7‰ V-PDB which are lower than those of reference and host rock
values.
The $\delta^{13}$C values of all dolomite types (+0.6 to +3.4‰) fall within the range of host rocks
and Jurassic marine limestones (Veizer et al., 1999). The $\delta^{18}$O shows a wider range of values,
somehow overlapping but also lower than those of host rocks (-4.5 to -0.9‰ V-PDB) and the
presumable Lower Jurassic marine dolomites. The majority of values plot between -3.5
and -1.5‰ V-PDB. The small size and overgrowth nature of certain dolomite types (e.g. D2 and



D5) limits their proper isolation for geochemical analyses. Only one sample from D1 dolomite
could be measured for $\delta^{13}$C and $\delta^{18}$O values, showing +2.5 and -1.9‰ V-PDB, respectively. The
$\delta^{13}$C and $\delta^{18}$O values of D3 dolomite range from +2.0 to +2.6‰ and -2.8 to -1.9‰ V-PDB,
respectively, with values lower than those of the host rock.
D4 dolomite has $\delta^{13}$C values between +2.4 and +2.5‰, and $\delta^{18}$O values of -3.0 to -2.5‰
V-PDB. The $\delta^{13}$C and $\delta^{18}$O values of CV2 are +1.2 to +3.1‰ and -1.7 to -1.7‰ V-PDB,
respectively. The $\delta^{13}$C values of CV3 are between +0.5 and +2.4‰, and the $\delta^{18}$O values cover a
range of -2.2 to 0.0‰ V-PDB. The $\delta^{13}$C and $\delta^{18}$O values of CV4 are +3.8 to +4.9‰ and -9.4
to -9.1‰ V-PDB, respectively. The $\delta^{13}$C values are slightly higher but the $\delta^{18}$O values are
considerably lower compared to preceding calcite generations and the measured values from host
rocks.
### 4.3.2 $^{87}$Sr/ $^{86}$Sr ratios
Samples from host rocks (i.e. Calcare Massiccio and Corniola Formations), dolomites
(D1, D3 and D4) and the Scaglia Formation in juxtaposition with the dolostones were analyzed
for their $^{87}$Sr/$^{86}$Sr isotopic ratios. The obtained ratios versus $\delta^{18}$O values of the analyzed samples
are shown in Fig. 12C. The $^{87}$Sr/$^{86}$Sr ratios obtained from the Calcare Massiccio and Corniola
limestones are 0.70766 and 0.70725 (n = 2), respectively, which is in agreement with the values
of the Lower Jurassic marine carbonates (0.70704-0.70768) reported by McArthur et al. (2012).
CV1 show a value equal to 0.70773.
All the dolomite types display higher $^{87}$Sr/$^{86}$Sr ratios when compared to the host rocks
and reference values of the Lower Jurassic marine carbonates. D1 (replacive) and D4 cements
show a comparable narrow range with values between 0.70784 and 0.70790, respectively. While,
the two D3 samples (replacive and cement) display higher $^{87}$Sr/$^{86}$Sr ratios (0.70858 and 0.70963,
respectively). The $^{87}$Sr/$^{86}$Sr ratios obtained for dolomites do not show co-variation with
corresponding $\delta^{18}$O values. The radiogenic Sr analysis was not performed on D2 and D5 since
the physical mixing with other dolomite types could not be avoided.
The $^{87}$Sr/$^{86}$Sr ratios of the two samples of Scaglia Formation are 0.70784 to 0.70790. The
CV2 veins in Scaglia Formation show comparable ratios of 0.70779 and 0.70787. These values
fit within the limits of values assigned by McArthur et al. (2012) for the Cenomanian-Bartonian
(Scaglia age) marine carbonates (0.70730-0.70790).



### 4.4 Fluid inclusion microthermometry


The overview of microthermometry measurements is given in Table 1 and Figs. 13A to
C. On the basis of optical microscopy analysis of wafers, D1 contain dominantly mono-phase
aqueous inclusions with sizes greater than 5 μm. It is common for small inclusions (< 3 μm) to
remain mono-phase all liquid at room temperature due to their metastability (Goldstein and
Reynolds, 1994). Thus, to eliminate the possible role of metastability, the samples were placed in
a freezer for several days following the procedures described in detail by Goldstein and Reynolds
(1994). All liquid inclusions remained unchanged and no vapor bubble was developed within
them, which discards the metastability effect.
In order to properly observe the phase transitions in the all liquid inclusions, they were
rapidly heated up to ~ 200°C to stretch and nucleate a bubble at room temperature (Goldstein,
1990). All the inclusions froze at -65 to -49°C. The first melting (Te) was detected between -22
to -19.3°C. The final ice melting (Tm) appeared at temperatures between -7.7 and -2°C.
Applying Bodnar´s (1993) equation, the obtained final melting temperatures correspond to
salinity ranges of 3.5 to 11.3 eq. wt. % NaCl.
D2 is characterized by the presence of mono-phase and infrequent two-phase inclusions
generally within their growth zones. The homogenization temperature of two-phase inclusions
varies between 58 and 71°C. Upon cooling, a complete freezing of the fluid phase is reached
at -56 to -40°C. The first ice melting temperature was distinguished at -22°C. The final ice
melting temperatures fall within -17.5 and -5°C, corresponding to salinities between 7.9 and
20.5 eq. wt. % NaCl.
D3 is commonly inclusion poor. The measureable inclusions were detected and examined
only in saddle dolomite crystals. These crystals contain only two-phase aqueous inclusions. Their
homogenization temperatures are within the narrow range of 70 to 73°C. The complete freezing
and first ice melting temperatures could not be distinguished but the final ice melting
temperature occurred at temperatures between -13 and -6°C equal to salinity ranges of 9.2 to
16.9 eq. wt.% NaCl. The first melting temperatures of fluid inclusions in D1, D2 and D3 were
about -21°C, suggesting a $H_2O$-NaCl fluid system.
D4 contains only two-phase aqueous inclusions. The homogenization temperatures in D4
vary between 79 and 105°C. Complete freezing of inclusions occurred at temperatures
between -86 and -54 °C. The first ice melting was detected at -35 to -40°C indicating the



possible presence of divalent cations such as $Ca^{2+}$ and/or $Mg^{2+}$ in the fluids (Shepherd et al.,
1985; Goldstein and Reynolds, 1994). The final ice melting temperatures fall within a range
of -15 and -9°C corresponding to salinities of 12.8 to 18.6 eq. wt. % NaCl. A couple of
inclusions show homogenization temperatures exceeding 120°C with salinities higher than
20 eq. wt. % NaCl. The inconsistent homogenization temperatures and salinities obtained for
these fluid inclusions, within the framework of an individual fluid inclusion assemblage (FIA)
described by Goldstein and Reynolds (1994), indicate possible re-equilibration of these
inclusions and thus are not used in the interpretations.
The obtained homogenization temperatures in all fluid inclusion assemblages indicate the
minimum temperatures at which the fluids could have been trapped (Goldstein and Reynolds,
1994). No correction was made for pressure effects on entrapment temperatures since no data
regarding the exact depth and pressure of entrapment are available. In absence of independent
thermal indicators such as Conodont Alteration Index (CIA) and Vitrinite Reflectance (VR), the
accuracy of pressure correction cannot be well constrained (Slobodník et al, 2006), and thus no
correction was made for pressure effects on homogenization temperatures.
No measurable fluid inclusion could be identified in CV1 and CV2 due to intense
deformation twinning. CV3 and CV4 contain only primary mono-phase aqueous inclusions,
indicating an entrapment temperature of below about 40-50°C (Goldstein and Reynolds, 1994).
A complete freezing of the inclusions in CV3 occurred at temperatures between -40 and -52.5°C.
The first melting temperature was detected at about -21 to -22°C, suggesting a $H_2O$-NaCl
composition. The final melting temperatures range between -6.4 and -2.7°C, corresponding to
salinities between 9.7 and 4.5 eq. wt. % NaCl. The majority of the values cluster between 7.8 and
5 eq. wt. % NaCl.
The complete freezing temperatures of the inclusions in CV4 fall within -46 and -35.5°C.
The first melting temperature could not be determined with confidence but the final melting
temperatures were reached at about -0.1 to -1.8°C, corresponding to salinities of 0.17 to
3.0 eq. wt. % NaCl.
**5 Discussion**
**5.1 Stable and radiogenic isotopic composition of the parental fluids**
The $\delta^{13}C$ values of all dolomite types mimic the range of host rock and Jurassic marine
limestones and, consequently, they can be interpreted as largely rock-buffered. Their $\delta^{18}O$ values





are partly comparable to those of their respective host rocks as well as Jurassic marine reference
values but more depleted when compared to the presumable Jurassic marine dolomites. The
relatively depleted $\delta^{18}O_{dolomite}$ values could indicate the contribution of heated fluids in
dolomitization process, although it could also relate to recrystallization of a precursor dolomite
by fluids at higher temperature or $^{18}O$-depleted (Land, 1980; 1985). The absence of distinctive
textural evidence in the analyzed samples such as enlarged crystal size and/or systematic mottled
cathodoluminescence pattern, and their co-variation with $\delta^{18}O$ values do not confirm
recrystallization (Mazzullo, 1992 and ref. therein). Nevertheless, special care was taken to avoid
the samples that occasionally displayed scattered mottled luminescence.
The oxygen isotope fractionation relation between water and dolomite (Land, 1983) was
used to determine the most plausible parental fluids. In order to avoid erroneous results due to
rock-buffered $\delta^{18}O$ values, only the $\delta^{18}O$ values of dolomite cements, especially from the bed
parallel veins containing D4 were used. These values may provide the closest approximation to
the $\delta^{18}O$ signature of the parental fluids (Barker and Cox, 2011). Accordingly, a $\delta^{18}O$ value of $\approx$
+2.5 to +4‰ V-SMOW was calculated for D3, while this values increase to $\approx$ +5 to +7.5‰
V-SMOW for D4 (Fig. 13D). The calculated compositions of the potential parental fluids are
progressively higher The higher $\delta^{18}O$ composition of the dolomitizing fluids relative to the
Mesozoic seawater, which is estimated at -1.2 to -1‰ V-SMOW (Shackleton and Kennett, 1975;
Marshall, 1992; Saelen et al., 1996), is compatible with fluids derived from or that had interacted
with siliciclastics, crystalline basement (Taylor, 1997) and/or evaporite-derived brines.
The $^{87}Sr/^{86}Sr$ ratios obtained for all dolomite types are higher than the Lower Jurassic
marine carbonate values (0.70704-0.70768; McArthur et al., 2012). Since marine carbonates
have very low rubidium (Rb) concentrations they produce negligible *in situ* radiogenic $^{87}Sr$ after
their deposition (Stueber et al. 1972; Burke et al. 1982). Therefore, the higher $^{87}Sr/^{86}Sr$ ratios can
be explained by the contribution of fluids originated or interacted with potassium rich
siliciclastics (K-feldspars), crystalline basement and/or stratigraphic levels with higher $^{87}Sr/^{86}Sr$
ratios (Emery and Robinson 1993; Banner, 2004). Taking into account that the Upper Triassic
Burano Formation underlying the studied intervals has $^{87}Sr/^{86}Sr$ ratios between 0.70774 and
0.70794 (Boschetti et al., 2005), the $^{87}Sr/^{86}Sr$ ratios (D1 and D4) can partially be explained by
their contribution. However, this contribution cannot justify much higher $^{87}Sr/^{86}Sr$ ratios
recorded in D3, being higher than values reported for Phanerozoic seawater (McArthur et al.,

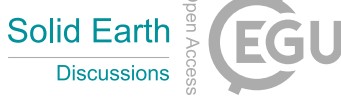


2012), and the values recorded in adjacent basinal deposits (i.e. Corniola and Scaglia
Formations). Therefore, parental fluids most likely originated from or had interacted with the
siliciclastics underlying the Burano Formation (Verrucano Formation), if present, and/or with the
crystalline basement with common elevated $^{87}Sr/^{86}Sr$ ratios (0.71500-0.72650; Del Moro et al.,

1982).

CV1 is characterized by $\delta^{13}C$ and $\delta^{18}O$ values lower than the host limestones (i.e. Calcare

Massiccio), while its $^{87}Sr/^{86}Sr$ ratio is comparable to them. The salinity and composition of the
parental fluids cannot be inferred here since no measurable fluid inclusions were found within
this cement. The $^{87}Sr/^{86}Sr$ ratio being within the range of the corresponding host rocks and the
reference values, points to a rock-buffered system for $^{87}Sr/^{86}Sr$.

The $\delta^{13}C$ and $\delta^{18}O$ values obtained for CV2, as well as $^{87}Sr/^{86}Sr$ ratios, fall within the

range of the Scaglia host rocks, thus reflecting their rock-buffered nature. This interpretation is
further supported by the comparable luminescence characteristics of CV2 with that of encasing
Scaglia host rocks. The fluids from which CV2 calcite precipitated were most likely derived
from carbonate dissolution during pressure-solution and stylolitization.

CV3 is characterized by $\delta^{13}C$ values within the Jurassic marine values but are generally

lower than the host rocks, while their $\delta^{18}O$ values partially overlap both the hosting limestones
and dolostones. Microthermometry of fluid inclusions revealed only mono-phase aqueous
inclusions and thus precipitation at relatively low temperature ($\leq$ 40-50°C) with moderate
salinity (4.5-9.7 eq. wt. % NaCl). Such levels of salinity can be assigned to evaporated seawater,
residual brines or fluids derived from evaporite dissolution, and thus makes it difficult here to
interpret their exact origin with the available data.

CV4 is the latest calcite phase, and records the $\delta^{13}C$ and $\delta^{18}O$ values, respectively

enriched and significantly depleted when compared to their hosting rocks and preceding
diagenetic products. Generally, the enrichment of $^{13}C$ could suggest $CO_2$ outgassing due to
evaporation (Friedman, 1970; Hendry et al., 2015) or bacterial fermentation (methanogenesis) of
organic matter (Hudson, 1977) in low temperature diagenetic environments. The homogenization
temperature of CV4 being below about 40-50°C could support any of these processes. Their low
$\delta^{18}O$ values and fluid inclusions with salinities comparable to, but also significantly lower than,
seawater reflect the contribution of meteoric fluids during precipitation of this calcite.
**5.2 Origin of the dolomitizing fluids**





The contribution of brines that derived from highly evaporated seawater or evaporites is
suggested by the elevated salinity values obtained from microthermometry of the fluid inclusions
(3.5 to 20.5 eq. wt. % NaCl). Accordingly, two sources that could potentially provide such fluids
can be proposed: 1) fluids related to the Late Messinian evaporites, associated with the overlying
Laga Formation, deposited during the Upper Miocene time, and their possible downward
percolation through fault zones by density driven flow and/or seismic pumping mechanisms
(Sibson, 1981; McCaig, 1988, 1990); or their tectonic involvement into the Apenninic thrust
wedge during its propagation (underthrusting; Lobato et al., 1983); and 2) fluids related to the
underlying décollement horizon of Burano evaporites (Upper Triassic) and their upward flow
through fault zones during development of the Montagna dei Fiori Anticline. The first scenario is
valid if the dolomitization would have occurred only from the Upper Miocene time onwards.
Several researchers (e.g. Vai and Ricci Lucchi, 1977; Bassetti et al., 1998; Roveri et al. 2001)
have shown that the occurrence of primary shallow-water evaporites, which were dominantly
gypsum, was limited to the western and central parts of the northern Apennines consisting of
thrust-top marginal basins. In contrast, evaporites never precipitated in parts of the central
Apennines including the Montagna dei Fiori region (Marche area) (Roveri et al. 2001). Hence,
the evaporitic horizons existing within the Laga Formation corresponds to resedimentation
(gypsum debris) of those previously precipitated in the marginal basins. This interpretation
makes the Messinian evaporites an unlikely source of Mg-rich brines. Moreover, taking into
account that the maximum burial related temperature of the Calcare Massiccio Formation did not
exceed 80°C in the Montagna dei Fiori region (Ronchi et al., 2003), it's not likely that the
downward percolation of relatively low-temperature brines derived from the Messinian
evaporites, located at the higher stratigraphic levels, could reach or exceed the high temperatures
recorded in fluid inclusions of the dolomites in the Calcare Massiccio Formation (D4; up to
105°C), given that the homogenization temperatures reflect the minimum entrapment
temperatures (Goldstein and Reynolds, 1994). Deep circulation of these brines, if existed, can
also be excluded by their limited tectonic involvement within the thrust wedge being confined
merely to the off shore wards of the Montagna dei Fiori region (Artoni, 2013).
Accordingly, the Upper Triassic Burano Formation appears as the most plausible source
for the high salinity brines recorded in fluid inclusions, and likewise, the Mg-rich fluids could
have been originated from post-evaporite brines associated with them (Carpenter, 1978;





McCaffrey et al., 1987). The fluctuations in salinity may argue for different degrees of
contribution of pore waters of lower salinity (e.g. marine or meteoric).

**5.3 Timing and structural controls on the evolution of parental fluids**

A generalized paragenesis and the relative chronology of dolomitization in relation to the
structural evolution of the Montagna dei Fiori Anticline are illustrated in Figs. 14 and 15. The
paragenesis is constructed on the basis of direct evidences recorded during observations at
outcrop scale and microscopic observations (e.g. cross-cutting relationships between diagenetic
phases, stylolites, fractures and other structural kinematics), and indirect evidences (e.g. regional
geodynamics and burial history).
The occurrence of micritic envelopes and fibrous calcite cements (FC), in grain supported
stratigraphic levels of the Calcare Massiccio Formation, is interpreted to be of eogenetic origin
(i.e. marine phreatic diagenesis; Moore, 1989), reflecting an early diagenesis shortly after
deposition. The well-developed dull and orange concentric cathodoluminescence pattern of the
succeeding mosaic calcite cement (MC) suggests a progressive shift to more reducing conditions
during precipitation in a phreatic diagenetic environment (as shown in Li et al., 2017). High
amplitude bed parallel stylolites postdate both cements, which confirm their precipitation before
significant burial. The observations made here are in agreement with earlier work by Giacometti
and Ronchi (2000), interpreting that the Calcare Massiccio Formation was cemented during the
early diagenetic stages.
D1, CV1 and D2 are postdated by well-developed, high amplitude bed-parallel stylolites.
Presence of D1 and CV1 in bed-perpendicular veins typically abutted by these stylolites (see
Figs. 6E to H) support the interpretation that the first dolomitization event (D1 and D2) took
place before significant burial and stylolite development, being the latter and bed-perpendicular
veins dynamically compatible within the same stress field characterized by a vertical, load-
related maximum principal axis of the stress ellipsoid. The dominantly mono-phase fluid
inclusions within D1 and D2 are in agreement with precipitation temperatures below about
40-50°C, suggesting a relatively shallow to intermediate burial environment and hence
supporting a pre-Apenninic orogeny age of precipitation from a mix of formational and extra-
formational fluids with elevated $^{87}Sr/^{86}Sr$ ratios. The distribution of D1 and D2 nearby the
rifting-related ~ N-S and E-W striking extensional faults and even their displacement along them
(Fig. 2A, e.g. site 1), point to the possible contribution of these faults in occurrence of D1 and



D2. These faults dominantly affect the Jurassic rocks older than the Maiolica Formation which is
attributed to the post-rift deposits, therefore suggesting a pre-Maiolica age for these dolomite
types. Although, an absolute age cannot be provided, based on the evidence discussed above, the
circulation of Mg-rich fluids during this dolomitization event was most likely controlled by
rifting-related Jurassic extensional fault zones cutting through the crystalline basement. Taking
into account that D1 and D2 are the volumetrically more relevant dolomites within the studied
intervals, and assuming the likely role of syn-rift extensional faults (Early to Late Jurassic) in
their precipitation, a dominantly syn-rift dolomitization process is proposed for the dolostones in
the Montagna dei Fiori Anticline.
D3 and D4 both record elevated $^{87}Sr/^{86}Sr$ ratios which accounts for their fault-controlled
origin. However, their occurrence at the top of the Calcare Massiccio and overlaying Bugarone
Formation (Corano Quarry site) which is < 1 m thick in Montagna dei Fiori region, and is
marked as the final rift deposit (Cardello and Doglioni, 2015) discards a syn-rift origin for these
dolomites. Moreover, D3 and D4 postdate the development of high amplitude bed parallel
stylolites. The formation of stylolites requires an approximate overburden of 600 to 1500 m
(Lind, 1993; Machel, 1999; Mountjoy et al., 1999; Schulz et al., 2016), corresponding to a late to
post-Maiolica deposition time (Early Cretaceous time onwards). The presence of D3 and D4
dolomites in bed parallel fractures and shear veins (D4) suggests their association with
contractional deformations, i.e. the most likely tectonic regime for explaining bed-perpendicular
dilation. Therefore, the volumetrically minor second stage of dolomite precipitation may
possibly be related to the Late- to post-Miocene compressional tectonics recorded in this region
(e.g. Mazzoli et al., 2002; Artoni, 2013; Storti et al., 2018).
Dolostones containing D3 and D4 appear commonly as clast-supported breccias along
fault zones pertaining to the Montagna dei Fiori Fault, then overprinted by fault-parallel
stylolites. Accordingly, the occurrence of these dolomites was probably synchronous with the
incipient stages of fault development, predating fault buttressing (Storti et al., 2018).
Homogenization temperatures recorded in D4 (up to 105°C), much higher than the maximum
temperatures recorded in the host rocks (below about 80°C; Ronchi et al., 2003), suggest
hydrothermal fluid circulation. The development of the Montagna dei Fiori Anticline at the toe
of the Late Miocene Central Apennines thrust wedge could have favored the forelandward
migration of hydrothermal fluids expelled from the more internal regions of the belt, similarly to



what has been proposed for the Rocky Mountains foreland (i.e. squeegee flow model; Machel and Cavell (1999). Such a migration may have possibly favored the precipitation of D4 in bed parallel veins, generally considered as evidence for syn-compressional fluid overpressure (Sibson, 2001; Hiemstra and Goldstein, 2015). The presence of D5 only within the damage zone of the Montagna dei Fiori Fault, postdating dolostone brecciation and, in places, cementing breccia fragments, may suggest that D5 dolomite precipitation was associated with the late stage evolution of the Montagna dei Fiori Fault, predating late stage calcite precipitation.

The presence of several generations of bed perpendicular stylolites bounding and intersecting CV2 veins, supports that late stage calcite cements precipitated closely associated with the deformation history of the Scaglia Formation in the hanging wall of the Montagna dei Fiori Fault, during buttressing against Calcare Massiccio and Corniola Formations in the footwall, and related with the positive inversion event induced by thrust-sheet stacking at depth (Storti et al., 2018). Precipitation of CV3 and CV4 in interpreted to have occurred during uplift and cooling as revealed by their relatively low homogenization temperatures ($\leq$ 40-50°C). Deformation twining is either absent or weakly developed, reflecting the lack of significant tectonic deformation after calcite precipitation. These cements postdate the dolomitization events, high amplitude bed perpendicular and parallel stylolites, and are precipitated as cements bounding the breccia fragments within the damage zone of the Montagna dei Fiori Fault. Salinities calculated from their fluid inclusions, particularly in CV4 suggests precipitation from meteoric waters, which should have been favored during the late evolutionary stages of antiformal stacking beneath the Montagna dei Fiori Anticline, and eventual late extensional slip along the Montagna dei Fiori Fault (Storti et al., 2018). The results obtained in this study are in relative agreement with the earlier work by Ronchi et al. (2003) and Murgia et al. (2004) in the Central Apennines, assigning dolomitization phases to the pre- and syn-orogenic deformations, although they did not specify the direct relation between the structures and the different types of dolomite.

**6 Conclusions**

The Lower Jurassic limestones outcropping at the core of the Montagna dei Fiori Anticline (Central Apennines, Italy) are massively affected by dolomitization, in damage zones of the pre-orogenic faults inherited from the Tethyan rifting and the ones formed during the Apenninic orogeny. Cross-cutting relationships between deformation structures, and results from

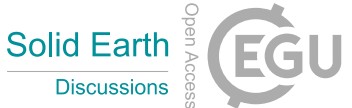

optical and cold cathodoluminescence petrography, fluid inclusion microthermometry, and isotope geochemistry, support the occurrence of two major dolomitization events. The first event is interpreted as developed during the late stages of Tethyan rifting in Jurassic and resulted in volumetrically significant dolostone geobodies. These dolostones are majorly matrix replacive and their precipitation initiated prior to the significant burial as reflected in their cross-cutting relationship with bed parallel stylolites, and by homogenization temperatures in fluid inclusions that are dominantly below about 40-50°C. The second dolomitization event corresponds to volumetrically less relevant replacive dolomite and dolomite cements occluding fractures. These dolomites precipitated during hydrothermal fluid circulation associated with contractional tectonics during the Apenninic orogeny, possibly at the onset of the growth of the Montagna dei Fiori Anticline (Late Miocene).

Dolomitizing fluids in both events were most likely sourced from evaporitic brines associated to the underlying Burano evaporites and their interaction with siliciclastics and/or the crystalline basement.

*Author contributions.* M. Mozafari participated in fieldwork, performed petrographic and microthermometric analyses, provided their interpretation, and wrote the manuscript; R. Swennen participated in fieldwork, discussed the results of the diagenetic study, and critically reviewed the manuscript; F. Balsamo contributed to collect and interpret structural data, discussed structural diagenesis data interpretation, and critically reviewed the manuscript; H. El Desouky collected $^{87}Sr/^{86}Sr$ data; F. Storti conceived the research, contributed to collect and interpret structural data, discussed structural diagenesis data interpretation, and critically reviewed the manuscript; C. Taberner participated in fieldwork, discussed the results of the diagenetic study and their framing into the proposed structural evolution, and critically reviewed the manuscript.

*Acknowledgments*. This research was performed by collaboration between Parma and KU Leuven universities in the framework of a research project (PT12432 and GFSTE 1100942) funded by Shell Global Solutions International (Carbonate Research Team, now Geology and New Reservoir Types Team). We thank E.M. Selmo (Parma University) and M. Joachimski (University of Erlangen, Germany) for the stable carbon and oxygen analysis. G. Davis (VU



Amsterdam, the Netherlands) is thanked for the strontium isotope analysis. A. Comelli and H.
Nijs are kindly thanked for the careful preparation of the wafers and thin sections. L. Barchi is
gratefully appreciated for his help in SEM analysis. We gratefully acknowledge A. Koopman for
the constructive discussions during field work.



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



**Table captions**

**Table. 1**. Stable carbon and oxygen isotopes, $^{87}Sr/^{86}Sr$ ratios, and fluid inclusion microthermometry data (not pressure corrected) of host rocks and diagenetic phases in the Montagna dei Fiori Anticline. Stable carbon and oxygen isotopes values are expressed in ‰ V-PDB and salinity values in eq. wt. % NaCl.

**Figure captions**

Fig. 1. A) Simplified regional map (modified after Ghisetti and Vezzani, 1997) showing the tectonic outlines of the Central Apennines and the study area (rectangle). B) Schematic geological map of the Montagna dei Fiori Anticline showing the distribution of dolostones (modified after Storti et al., 2017a). C) Lithostratigraphical column of the successions exposed in Montagna dei Fiori (modified after Mattei, 1987; Di Francesco et al, 2010; Storti et al., 2018). Lithologies are mentioned in the text. Note that the thickness of the not-outcropping formations (Triassic evaporites and the crystalline basement) is not to scale. D) Geological transect across present day Central Apennines and the Adriatic Sea (modified after Fantoni and Franciosi, 2010) with vertical exaggeration of 2:1. The dashed rectangle indicates the Montagna dei Fiori Anticline region.

Fig. 2. A, B) Geological map of the central sector of the Montagna dei Fiori Anticline, and cross-section oriented parallel (a-b) to the hinge line representing the tectono-stratigraphic architecture of the faulted anticline (modified after Storti et al., 2017a). The stereonets (Schmidt equal area projection lower hemisphere) provide the attitude of the extensional faults. The locations of the corresponding field sites are indicated by numbers.

Fig. 3. A) Field photograph showing the deformed Scaglia Formation in the hanging wall (HW) and brecciated, dolomitized Calcare Massiccio Formation in the footwall (FW) of the Montagna dei Fiori Fault. B) A hand specimen from the deformed Scaglia formation showing the intensity of the pressure solutions (TS) and their abutting relationship with calcite veins (CV2). C) A transmitted light photomicrograph of the dolomitized, brecciated Calcare Massiccio Formation. Note all the breccia fragments are composed of dolomite (D4 here).



Fig. 4. Field photographs (Corano Quarry) showing the field relations between dolostones, host limestones and the Montagna dei Fiori Fault: A) Panoramic view showing the spatial relationship between limestones and dolostones (orange) in the damage zone of the Montagna dei Fiori Fault (F). Note that the limestones and including dolostones of the Calcare Massiccio and Bugarone Formations on the footwall (FW) and marly limestones of the Scaglia Formation on the hangingwall (HW) are intensely deformed. B) Plan view of the Calcare Massiccio limestone in the footwall damage zone: intersected by calcite veins (CV1), dolomitized and affected by bed perpendicular stylolites (arrows). C) Distinct transition (dashed line) between dolomitized and undolomitized Calcare Massiccio limestone in the footwall damage zone.

Fig. 5. Field photograph (A) and a simplified sketch (B) of a dolomitic pocket within the folded Calcare Massiccio (grey color) and their relation with bed parallel stylolites (hammer is 40 cm long).

Fig. 6. A) Transmitted light image showing a micritic peloid rimmed by the fibrous cements (FC) which are followed by the mosaic cements (MC). B) Transmitted light image showing mosaic cements (MC) in a peloidal limestone over printed by high amplitude bed parallel stylolites (dotted white line). Note the core of some of the peloids is partially cemented as well. C, D) Respectively, transmitted light and corresponding cathodoluminescence image of FC and MC cements. E) Transmitted light photomicrograph showing D1 crystals rimming a fracture which is cemented by CV1. The fracture is in turn affected by a bed parallel stylolite. F) Cathodoluminescence image showing D1 scattered in the host rock and riming the fracture. G, H) Respectively, transmitted light and corresponding cathodoluminescence image showing part of a bed parallel stylolite (dotted white line) overprinting D1 and D2 crystals.

Fig. 7. A, B) Photomicrographs of respectively, transmitted light and corresponding cathodoluminescence image showing the zoned rhombs of D2 with the remnants of D1 preserved in their cloudy core. The pore space is occluded by D4. C, D) D3 cementing angular breccia fragments of the Bugarone Formation in the damage zone of the Montagna dei Fiori Fault in the Corano Quarry site. Note the breccia is overprinted by a fault parallel bed perpendicular stylolite. E, F) Photomicrographs of respectively, transmitted light and corresponding



cathodoluminescence image showing the euhedral to subhedral crystals of D3 developing a
bright subzone and rim. G, H) D3 with a saddle crystal outline (SD) postdating calcite cements
(MC) and a zoned D2 crystal. The saddle morphology is outlined by a dotted white line.

Fig. 8. Photomicrographs of respectively, transmitted light and corresponding
cathodoluminescence image of dolomite types: A, B) The cross-cutting relationship between D3
and D4. Note the presence of D3 within the core of dolomite crystals overgrown by D4. C, D)
Successions of dolomite types. Note the green CL color of D4 crystals. Typically, luminescent
dolomites are known to show yellow, orange to red colors (Machel et al., 1991). Green
luminescence in carbonates including dolomite have been attributed by a number of researchers
to the incorporation of three valent rare earth elements (REE) such as $Dy^{3+}$ and $U^{3+}$ as
luminescence activators within their crystal lattice (Luczaj and Goldstein, 2000). Another
possibility is the emplacement of $Mn^{2+}$, with yellow luminescence, in $Ca^{2+}$ sites with blue
luminescence in the dolomite crystal lattice instead of preferential incorporation in the $Mg^{2+}$ site
(Sommer, 1972b; Amieux, 1982; Walker et al., 1989; Habermann et al., 1999). Accordingly,
non-stoichiometric, Ca-rich and poorly ordered dolomites may favor $Mn^{+2}$ incorporation into
their $Ca^{2+}$ site. E, F) Vuggy porosity rimmed by D4 (green CL). Note the porosity is filled with
fine dolomite rhombs including traces of D2 in their core and D4 overgrowths.

Fig. 9. Photomicrographs showing respectively, transmitted light and corresponding
cathodoluminescence image of D4 and D5 in relation to stylolites and fracturing: A, B) D4,
exploiting a bed parallel stylolite that crossed-cuts D1 and D2. C, D) A sub-horizontal fracture
cemented by D4. E, F) D5 microveins (arrows) intersecting all the predating dolomite types in
the footwall brecciated zone of the Montagna dei Fiori Fault.

Fig. 10. Field photographs showing the major calcite vein settings observed in Montagna dei
Fiori: A) Cross-sectional view of bed normal Calcite vein 1 (CV1) abutting bed parallel stylolites
in folded beds of the Calcare Massiccio Formation. B) Plan view of the Calcite vein 2 (CV2)
intensely affecting the deformed Scaglia (Rossa) Formation. C, D) Cross-sectional view of the
Scaglia Formation, intensely affected by pressure solution seams of tectonic origin crossed-over
by populations of bed-perpendicular Calcite veins (CV3) in en echelon extensional arrays.




Fig. 11. A) Cathodoluminescence and transmitted light (in set) image showing blocky to
elongated crystals of CV1 with zoned CL pattern. B) Transmitted light image showing intensely
twinned CV1 crystals overprinted by euhedral to subhedral crystals of D3. Photomicrographs of
respectively, transmitted light and corresponding cathodoluminescence image: C, D) CV2 in the
Scaglia Formation abutted by a bed perpendicular stylolite (indicated by white arrows and
dashed line). The crystals display blocky to fibrous morphologies, deformation twining, and a
similar orange luminescence pattern comparable with the adjacent host rock. E, F) CV3
cementing the breccia fragments in the damage zone of the Montagna dei Fiori Fault. The
crystals are blocky and show faint deformation twinning. They are brown-orange with distinct
darker luminescence sector zones. G, H) CV4 present as a cement within a polygonal pore space
rimmed by dolomite. Note the blocky crystals, absence of deformation twinning and distinct
concentric luminescence zonation pattern. CV4 is corroded and followed by a late telogenetic
calcite.

Fig. 12. A , B) Overview of the $\delta^{13}C$ and $\delta^{18}O$ values of dolomites (A) host rocks from Montagna
dei Fiori as well as calcite veins (B). The stable isotope value of Lower Jurassic marine
limestones based on Veizer et al. (1999) is indicated by a dashed rectangle in subset B. The $\delta^{18}O$
values of the marine dolomites are considered to be 3-4‰ V-PDB higher than those of marine
limestones (Land, 1980; Major et al., 1992; Horita, 2014). C) Cross-plot of $^{87}Sr/^{86}Sr$ ratios and
corresponding $\delta^{18}O$ values of host rocks, dolomites and calcite veins compared with Lower
Jurassic marine carbonates $^{87}Sr/^{86}Sr$ (dashed rectangle) framework reported by McArthur et al.

1187   (2012).


Fig. 13. Overview of microthermometry analysis of primary inclusions in Montagna dei Fiori: A)
Frequency distribution of the $Tm_{ice}$ (°C) in dolomite phases. B) Frequency distribution of the
Th (°C) in dolomite phases. C) Salinity (eq. wt. % NaCl) versus Th (°C) of dolomite and calcite
phases. D) Isotopic fractionation diagram from Land (1983) used to determine the isotopic
composition (‰ V-SMOW) of parental fluids in equilibrium with dolomites in Montagna dei
Fiori.

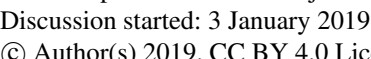
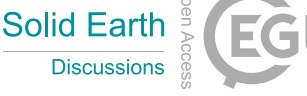

Fig. 14. A) Generalized paragenesis of diagenetic phases in relation to deformational stages and burial history of the Calcare Massiccio Formation in the Montagna dei Fiori Anticline. The deformational stages are from Storti et al. (2018), and the burial curve is based on Ronchi et al. (2003).

Fig. 15. Sketch showing the successive fault-related diagenetic phases, of most importantly dolomitization, recorded in the carbonate succession exposed at the core of the Montagna dei Fiori Anticline (not scaled). Different diagenetic phases are indicated with different colors. A) The first dolomitization event is pre-orogenic (syn-rift), triggered from the fluids channelized along Jurassic extensional faults. This event occurred during burial compaction and development of bed parallel stylolites (BS). It is represented by scattered dolomite rhombs (D1) followed by calcite cementation (CV1). The dolomitization continued with precipitation of larger crystals of D2. B) Second dolomitization event: syn-orogenic (early folding/ faulting) dolomitization from fluids that migrated from more internal regions of the thrust belt and were channelized along the basal detachment level into the fold core. This dolomitization event presents matrix replacive and cements displaying infrequent saddle outlines (SD) in pore spaces, within bed parallel veins and shear fractures. These dolostones postdate compaction but are affected by bed perpendicular stylolites (TS) generated by horizontal to sub-horizontal layer parallel shortening related to the growth of the Montagna dei Fiori Anticline. C) Extensional collapse of the anticline and development of the Montagna dei Fiori Fault, followed by buttressing of the Scaglia against Calcare Massiccio and Corniola Formations during positive inversion induced by continuing underthrusting at depth. Precipitation of D5 in micro-veins and cements in breccia zones, followed by late stage calcite cementation in the Montagna dei Fiori Fault damage zone (CV2, CV3 and CV4).



| | Stable isotopes | | Sr isotopes | Fluid inclusion microthermometry | |
| --- | --- | --- | --- | --- | --- |
| | $\delta^{13}C$ | $\delta^{18}O$ | $^{87}Sr/^{86}Sr$ | Th (°C) | Salinity |
| Calcare Massiccio Fm. | +2.4 to +3.1 | -1.6 to 0.0 | 0.70766 | - | - |
| Corniola Fm. | +2.0 to +2.5 | -3.1 to -1.4 | 0.70725 | - | - |
| Scaglia Fm. | +1.0 to +3.1 | -2.2 to -1.0 | 0.70784-0.70791 | - | - |
| D1 | +2.5 | -1.9 | 0.70789 | ≤ 40-50 | 3.5 to 11.3 |
| CV1 | +1.6 to +2.1 | -4.7 to -2.7 | 0.70773 | - | - |
| D2 | - | - | - | ≤ 40-50 to 71 | 7.9 to 20.5 |
| D3 | +2.0 to +2.6 | -2.8 to -1.9 | 0.70859-0.70964 | 70 to 73 | 9.2 to 16.9 |
| D4 | +2.4 to +2.5 | -3.0 to -2.5 | 0.70790 | 79 to 105 | 12.8 to 18.6 |
| CV2 | +1.2 to +3.1 | -1.7 to -1.6 | 0.70779 - 0.70787 | - | - |
| CV3 | +0.5 to +2.4 | -2.2 to 0.0 | - | ≤ 40-50 | 4.5 to 9.7 |
| CV4 | +3.8 to +4.9 | -9.4 to -9.1 | - | ≤ 40-50 | 0.17 to 3.0 |

Table. 1





Fig. 1



Fig. 2




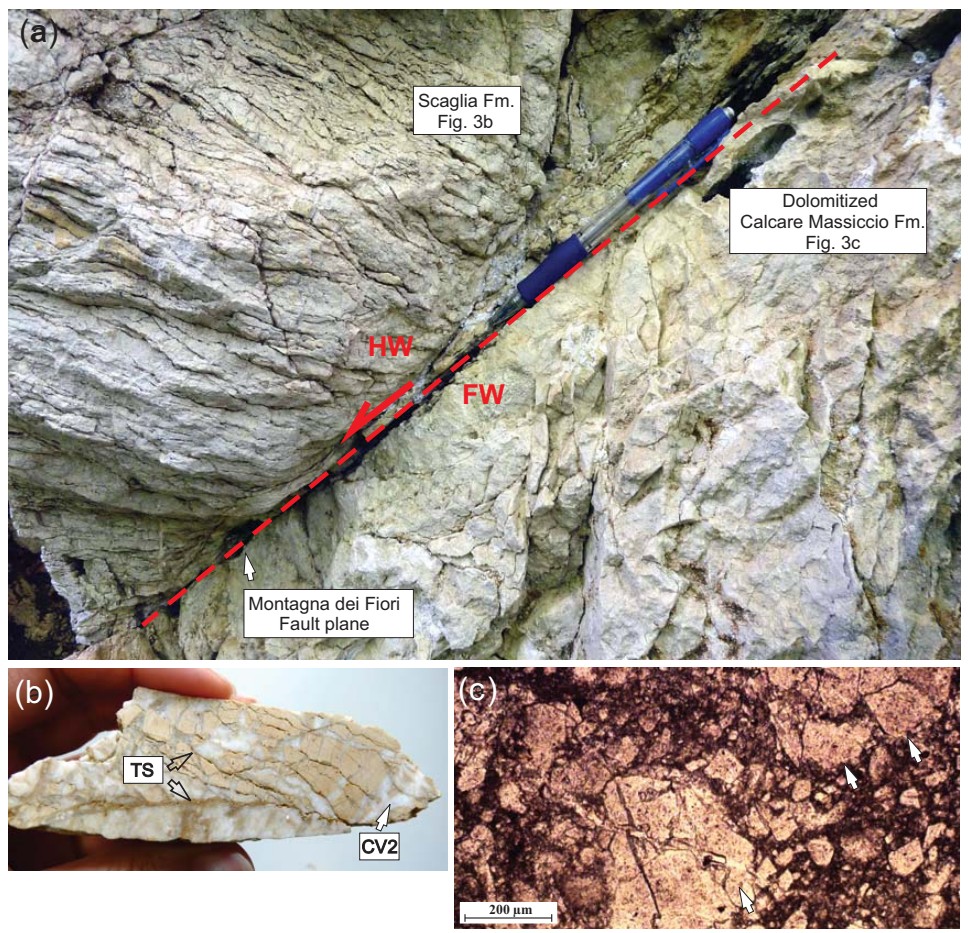

Fig. 3



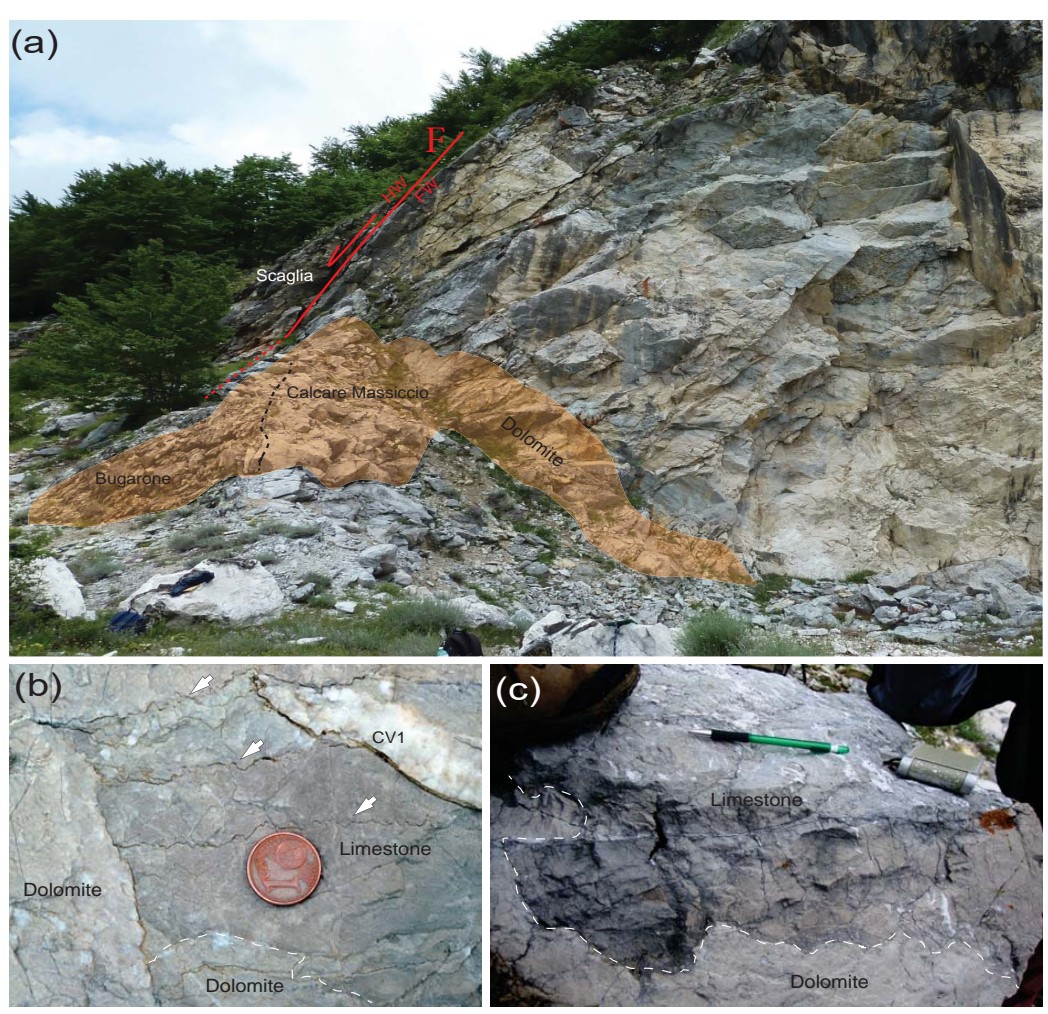

Fig. 4



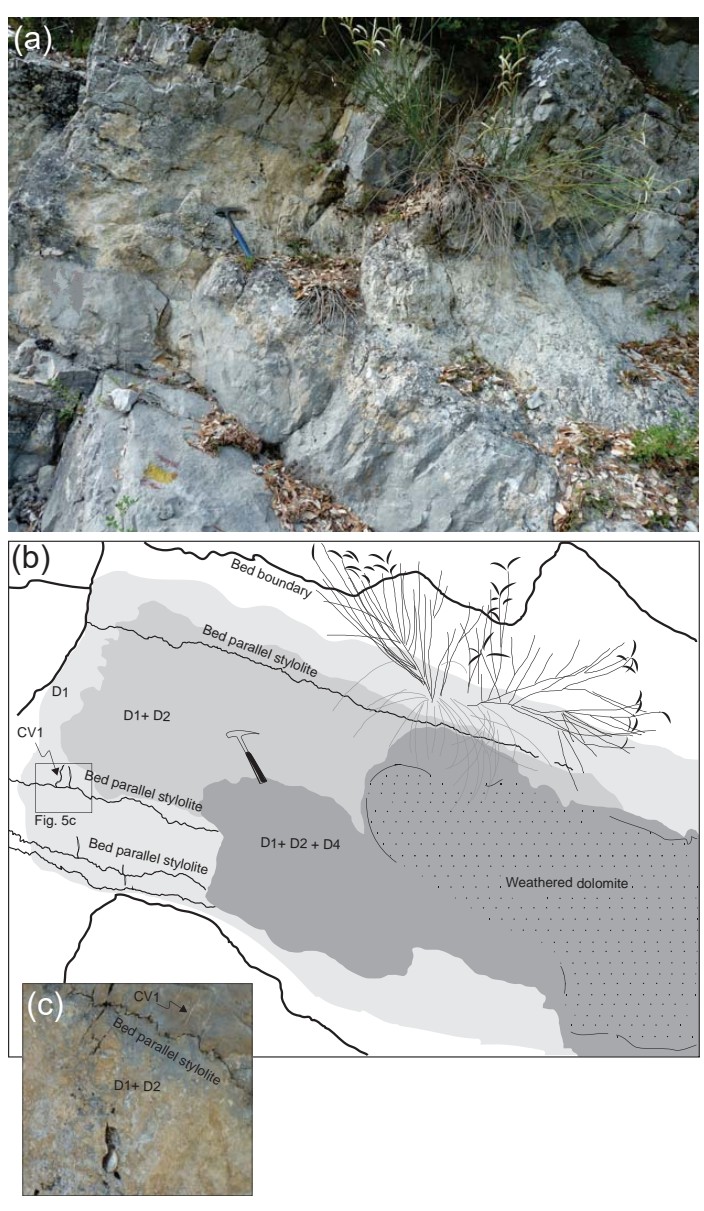

Fig. 5

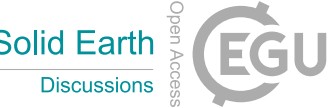



Fig 6





Fig. 7



Fig. 8





Fig. 9



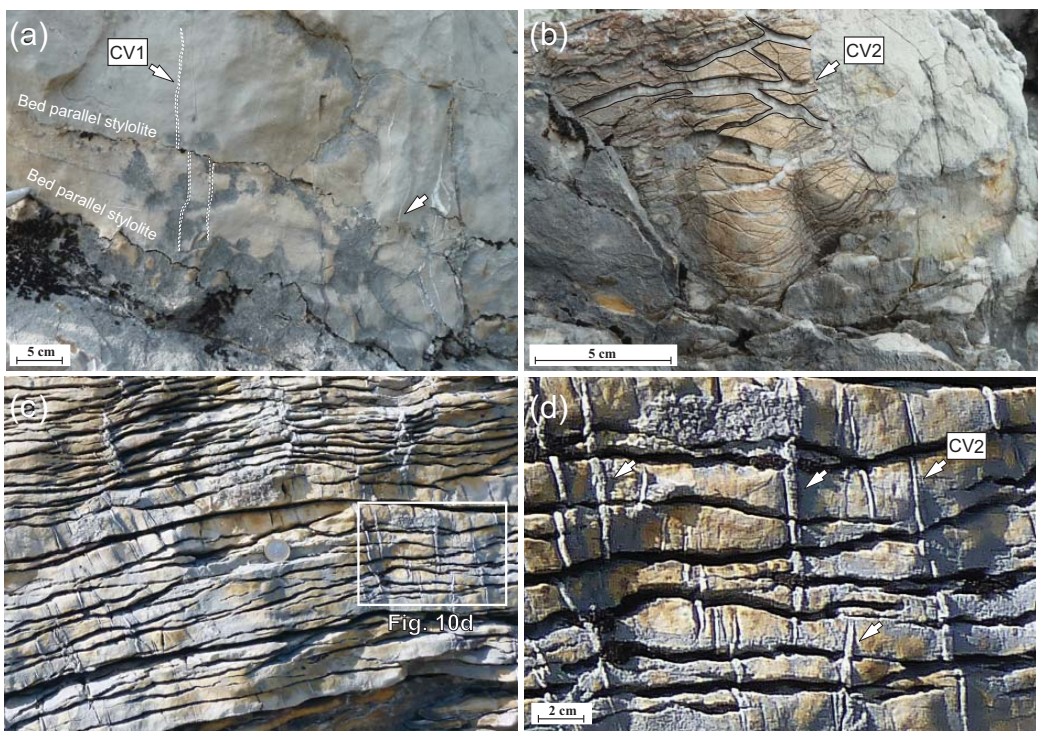

Fig. 10





Fig. 11



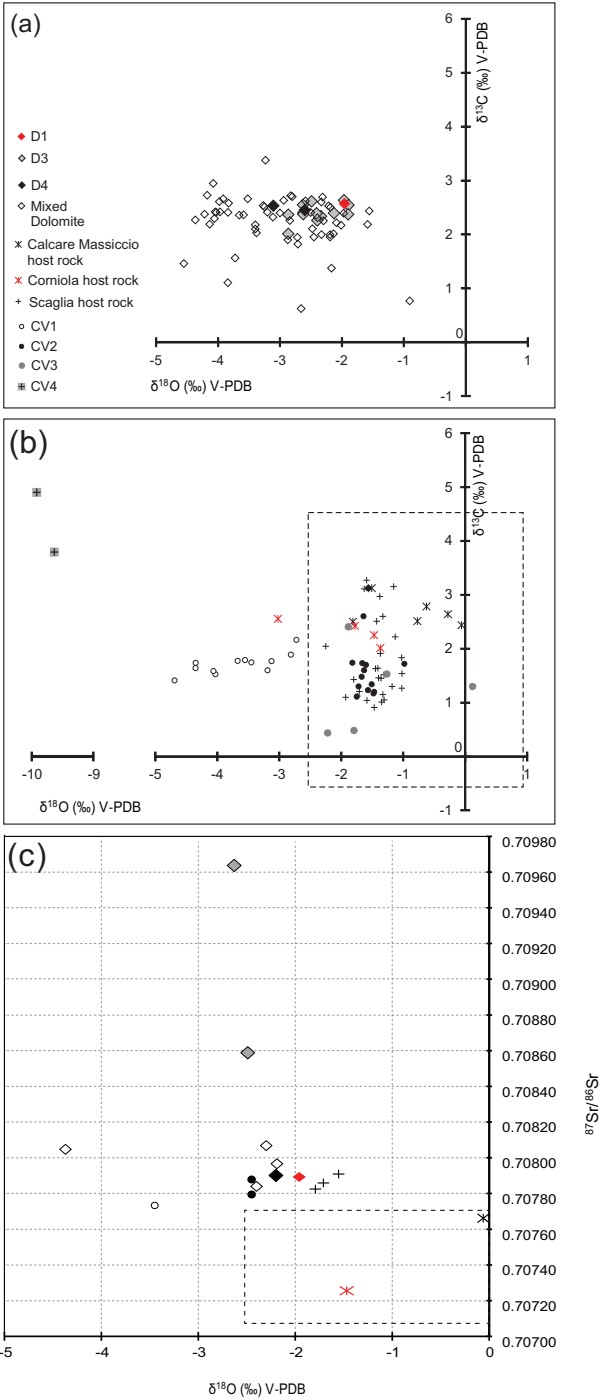

Fig. 12





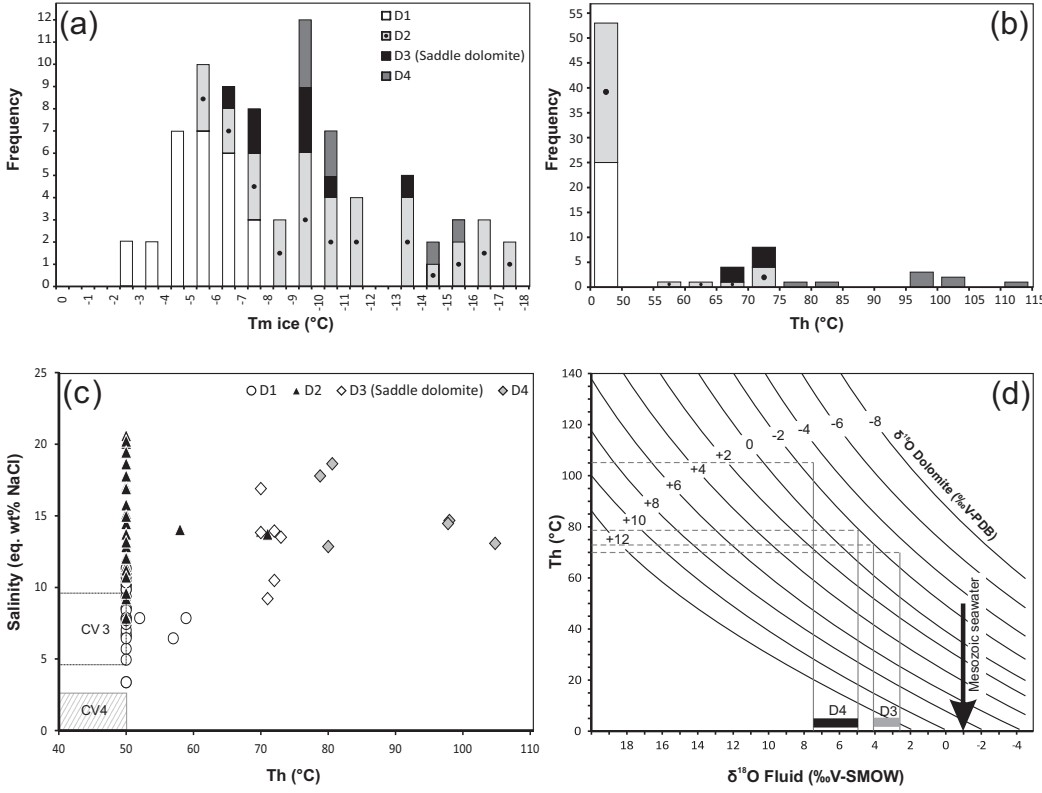

Fig. 13





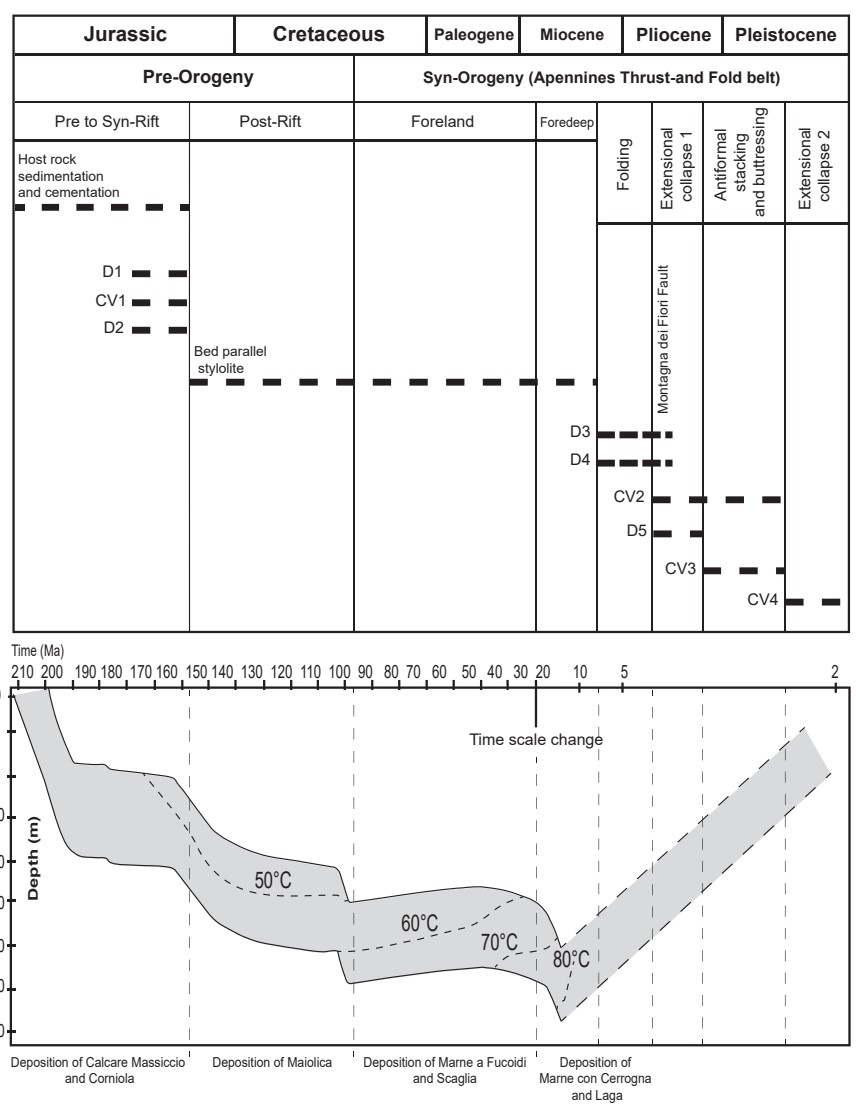

Fig. 14





Fig. 15