# Peer review of "Fault-controlled dolomitization in the Montagna dei Fiori Anticline (Central Apennines,"

_Solid Earth, 2018_

## Referee Comment (RC1) · Hendry (Referee) · 23 Jan 2019

**Review of "Fault-related dolomitization in the Montagna dei Fiori Anticline (Central Apennines, Italy): Record of a dominantly pre-orogenic fluid migration. Se-2018-136**

This manuscript presents field, petrographic and geochemical data from non-stratabound dolomites in a complex tectonic setting, and interprets their geofluid origin (parent fluids, timing) in the context of the tectonostratigraphic history.

It is coherent and logically organised. Aspects of the written English need minor improvement (punctuation, plurals, word order, etc); it will benefit from a final revision by a native English speaker.

The data are generally of good quality, and the interpretations are mostly justified from the results presented. In any study such as this, with limitations imposed by the ability to sample all the phases, there is necessarily some latitude or flexibility in the deductions that can be made. However, the authors do a good job of considering alternative possibilities for the fluid sources and timings.

My only issue with the paper is that the authors have not really considered whether there are wider implications or generic advances that can be made from the research. It presents itself as a case study, albeit one with a good integration of structural and diagenetic data. But what is the wider impact that will attract a non-specialist readership? Within the paper the authors all but admit that their findings are only modestly advanced from those of Ronchi and co-workers fifteen years ago (lines 638-642). I had hoped to see more progression in the science, and maybe the authors need to more thoroughly and critically evaluate the Ronchi model in the light of their new data. They could also work the structural data more – rather than just considering fault orientations and timings, what about the character and extent of the damage zones associated with different fault types / generations, and their relationship to the size and shape of dolomite bodies? What determines the lateral extent of dolomites? Is it other faults / fractures, or a gradual reaction front?

One generic aspect that the authors could address is implications for reservoir potential in analogues for this setting. The preponderance of planar dolomite is significant because planar dolomite is usually very beneficial for porperm (unlike many examples of hydrothermal dolomitization that feature tight nonplanar dolomites). Are there dolomitised plays in the Middle East that this study could be compared to (Zagros Mountains for example?), or maybe in Mexico?

Another factor of interest, largely by-passed in the text, is what drove the fluid circulation necessary to cause massive dolomitization when the low temperatures argue against a hydrothermal syn-rift system. Can the authors attempt a mass balance to estimate the order of magnitude fluid volumes needed? Maybe the dolomitization occurred in the down-flowing (cool) limb of a convection cell established on syn-rift faults that breached contemporary sea bed? Or does the dolomite zoning imply a pulsed fluid flow associated with strain cycling or seismic valving? I recommend the recent papers by Hollis and others on the Hammam Faraun fault and related syn-rift dolomitization. Are the D1-2 dolomites formed in a similar manner to this geologically younger example? Likewise, with the later dolomitization, which structures would have been open during compressional tectonics and able to serve as conduits for substantial fluid volumes?

If the authors address these issues their paper, which is already technically good, it will have much greater impact and interest across the sedimentary and structural geoscience community.

I have some more **specific comments** – these are tagged by line number or Figure number:

Line 43: The paper describes the role of evaporite-sourced fluids in the dolomitization process, but I am not sure that it illustrates a controlling role of evaporitic detachments. These may have influenced the tectonic development, but if it is believed that they directly controlled the dolomitization this needs to be specifically discussed later in the paper.

Note the abstract is quite long-winded. It would be good to make it more succinct and punchier.

Line 69: The Castel Manfrino Dolostones are not labelled in Fig. 1 or Fig. 2b.

Line 73, 76: Did Ronchi (2003) base her study on the mapping of Mattei (1987)? Maybe there needs to be a couple of sentences describing Ronchi's findings so that it can be more clearly shown that the understanding has moved on.

Line 79-83: This is very long-winded and vague. It either needs to be shortened or to include specific details.

Line 129: Given that the early dolomitization (D1-2) is later ascribed to the syn-rift stage, it would be useful to briefly describe the facies and architectural character of the syn-rift carbonates. For example, were they preferentially developed on footwall highs, in which case there was likely a juxtaposition of permeable high energy facies against faults that later hosted fluid flow?

Line 135-137: It may be a matter of debate, but the authors need to either provide the conflicting evidence or at least express a view and justify it.

Line 152: Can the Montagna dei Fiori fault be indicated / labelled on Fig. 1d?

Line 164: Ground truthing implies that the features had previously been mapped out using remote data. If so, this should be included in the methods.

Line 167: So far as I can see, the Sibley and Gregg (1987) terminology (planar-e, planar-s, nonplanar) is <not> used anywhere in the paper, so either it needs to be incorporated or this sentence should be removed.

Line 186: Can the reproducibility (±0.1‰) be smaller than the precision (±0.2‰)?

Line 224: Bugarone Formation is not shown on Fig. 1 or Fig 2.

Line 231: Is the wider distribution of dolomitized intervals related to the topography of the valley and the exposures? If not, what is the relationship?

Line 238: I suggest not using "overprinted", which implies the original fabric / lithology is lost. Why not just use "cross-cut"? (or even just "cut")

Line 258 and elsewhere: "Dull" is not a colour!

Line 261 onwards: There could be a bit more detail on the dolomite distribution and fabric with respect to host rock facies. Is it all texturally destructive, is there any textural or mineralogical selectivity, were grainy or muddy facies preferentially dolomitized (controls by permeability versus reactive surface area…….?)

Line 272: There is an issue because CV1, CV2 etc. are introduced before they have been defined and described. I suggest starting section 4.2 with a paragenetic summary to alleviate this problem.

Line 278: By using "frequently" the text suggests that sometimes (infrequently) D2 post-dates bed-parallel stylolites. Is that the case?

Line 284-285: This sentence needs a figure citation.

Line 297: Repetition of "euhedral to anhedral" (cf. line 295) – note this is not Sibley and Gregg terminology. Nor is "tightly packed texture" in line 279.

Line 305: I do not think one dolomite can "recrystallize" another. Recrystallization is a solid-state process that increased mineral stability. To demonstrate it might need data on the ordering, crystallinity and stoichiometry of D1/2 versus D4 (do the authors have any XRD data?). What is more likely is that D4 has locally replaced D1 and D2 by a dissolution-precipitation mechanism. However, the text lacks a clear description of the evidence for replacement. I recall papers by Mazzullo and by Machel that discuss this – it would be good to list the criteria for this case.

Line 330: In Fig. 11C, D the dolomite does not appear to be yellow-orange, it looks more like orange-brown.

Line 332: How wide was the extensional fault master plane? Please supply the range of widths (and lengths where possible) of the different vein generations.

Line 335-336: What is meant by "with no evidence of physical disruption"? Does it mean that CV3 always passively overgrows D5 in voids and never cuts it? If so, it is easier to say this.

Line 336: Translucent is not a colour.

Line 334: What colours are the zones?

Line 367-368: Rather than "the presumable Lower Jurassic marine dolomites" – which are hypothetical – it is better to say the values are lower than those expected for Lower Jurassic seawater dolomites.

Line 391: "While….." indicates there should be a second part to the sentence.

Line 396: What is the lithology of these samples?

Line 401: Please add a sentence or two on the fluid inclusion petrography and distribution – do inclusions follow growth zones or are they randomly distributed? Are they all primary or are some pseudosecondary? What are their shapes and what are the liquid:vapour ratios in the 2-phase examples? Also, in reporting the results for different cements please give the number of inclusions the ranges are based on (n=).

Line 409-410: What is the purpose of nucleating a bubble for measurement of freezing temperatures?

Line 477-478: This sentence appears out of place or at least needs clarification. More than two values are needed to demonstrate a progression.

Line 503-507: Yes, this makes sense if the veins are filling tension gashes associated with stylolites – such as system is likely to be buffered by the dissolving carbonate. Maybe make this point more explicitly, and contrast with vein types that were more extensive and would have allowed allochthonous fluids to pass through with minimal host-rock interaction.

Line 518: Hendry et al. (2015) did not discuss $^{13}C$ enrichment from $CO_2$ outgassing due to evaporation. They made the point that negative covariance in C and O isotopes within veins could be due to precipitation during $CO_2$ outgassing related to pressure changes.

Line 524-551: This is good but is a very long paragraph. Can it be made more succinct?

Line 550-551: The final sentence needs rewriting; what was confined, the thrust wedge or the fluids?

Line 556: In the preceding section there is very little mention of fluid mixing. Could the poorly correlated Th, salinity and stable isotopic data reflect precipitation from allochthonous fluids as they mixed with in situ fluids (and cooled)? Degrees of mixing (and of water-rock interaction) may have different from fault to fault – is it really likely that the hydrogeological systems was as simple as is being presented here?

Line 575-579: Please rewrite this sentence – it tries to say too many things at the same time.

Line 584: Doesn't the displacement of D1-2 on these faults indicate that the dolomite formed before faulting? What is the critical evidence that it is genuinely syn-rift?

Line 590: if D1-2 were related to basement-cutting faults, why are the Sr isotope values much less elevated than for D3 and D4?

Line 656-657: Please explain how hydrothermal fluids were able to circulate in the compressional tectonic regime – which structures were able to be in tension and therefore transmissive rather than sealing?

Line 1139: The cross-cutting relationship in Fig. 8a, b is not very evident.

Lines 1142-1148: Should this discussion be in the main text rather than in the caption?

Fig. 2b: It would help if the colours and ornaments matched Fig. 8a (e.g. Salinello Fm). The text size needs to be increased for better legibility. The stereonet data are good, but very little use is made of them in the text of the paper.

Fig. 5: Please increase the text size and make 5c larger – it is too small to see clearly.

Figs 6-9, 11: Some of the CL images could be a bit sharper and maybe with increased contrast to better discriminate the dolomite types.

Fig. 12: The symbols for D3 vs. CV3 and mixed dolomite vs. CV1 are too similar (especially given the small size). I am also not clear how Fig. 12b relates to Fig. 12a; maybe split the legend between the two plots according to what is in them – that might help.

Fig. 14: How were the burial temperatures in the burial history determined? What assumptions are they based on?

Fig. 15: I like this figure but I'm still not sure what the fluid flow pathway is in (b). Maybe a broader tectonic context diagram is needed showing expulsion of fluids from the foreland (if that is where they are coming from?).

I hope these comments are helpful, and I look forward to seeing the paper published in revised form.

Jim Hendry

Tullow Oil Ltd, Dublin

---

## Author Comment (AC2) · 22 Feb 2019

Dear Dr. Hendry, Thank you for the very constructive comments. They not only helped us to improve the quality of the manuscript but also our knowledge of dolomitizataion process. We have tried our best to address your suggestions in this new version of the manuscript. Regarding the advancement of our research in comparison with the work previously presented by Ronchi et al. (2003), the current study gives much more details about the dolomite characterization and their relation to the structural evolution of the anticline on the regional and local scale. Furthermore, the obtained geochemical

and microthermometry analyses do not confirm the role of marly or shaly basinal successions in providing the Mg-rich fluids during the first event of dolomitization (i.e. syn-rift), as proposed by Ronchi et al. (2003). We have tried to be modest in criticizing the latter authors limited research since the current research was build up on their findings. During our research, we also performed some other advance analyses such as clumped isotopes and U/Pb dating. However, the consecutive overgrowth pattern of dolomites and difficulties in isolating them to get enough and good quality samples increased the uncertainty in the results. Therefore, we decided not to include those data in the manuscript. In the study area, the structures and their relative chronology are very complicated. A comprehensive structural study on the evolution of the Montagna dei Fiori Anticline was performed parallel with the current study, and published by Storti et al (2018) in Tectonics. The target of the current study was to focus on dolomitization, and to use the structural model proposed by Storti et al (2018) to deduce the most likely timing for dolomitization. Another important question about the studied dolomites was the role of Scaglia Formation in providing the Mg-rich fluids during compression, because this formation is juxtaposed with the dolostones by the Montagna dei Fiori Fault. Our results do not support this hypothesis. Moreover, we show that the dolomitization predates the observed juxtaposition. The revised manuscript including track changes have been uploaded as a zip file. The resolution of figures is reduced to make a smaller pdf file. The original figures have much higher quality than the uploaded ones. As you recomended, the manuscript will be reviewed by a native speaker, but after receiving the second reviewers' comments. Best regards, Mozafari et al.

Please also note the supplement to this comment:
https://www.solid-earth-discuss.net/se-2018-136/se-2018-136-AC2-supplement.zip

---

## Referee Comment (RC2) · Jim Hendry (Referee) · 1 Mar 2019

Dear Mahteb,

I am satisfied that your revision has addressed the comments that I raised in my review. My only, slight, remaining issue is that I feel the hydrodynamic drive for fluid flow could be more rigorously addressed (especially in terms of fluid volumes required).

Otherwise I look forward to seeing you work published.

[Figure]

Kind regards,

Jim Hendry

---

## Referee Comment (RC3) · Jim Hendry (Referee) · 2 Mar 2019

Dear Mahtab,

There was a paper by Lynton Land in the mid 1980s that did a mass balance - it should be referenced in any of the more recent reviews. Try Warren's paper in Earth-Science Reviews 52 2000. 1–81. Or maybe one of Hans Machel's dolomitisation review papers may mention it.

Kind regards,

Jim

---

## Author Comment (AC3) · 2 Mar 2019

Dear Dr. Hendry, Thank you for your reply. Can you please introduce some papers regarding calculation of the fluid volume required for dolomitization? Much appreciated. Mozafari et al.

---

## Referee Comment (RC4) · Estibalitz Ukar (Referee) · 21 Mar 2019

This paper presents field, petrographic, isotopic, and fluid-inclusion thermometric and compositional analyses of the different dolomitization and calcite veining events that affected Lower Jurassic rocks within the Montagna dei Fiori Anticline. The authors conclude that dolomitizing fluids show evidence of interaction with underlying units and therefore infer that dolomitization was fault related and occurred in two main episodes, before and during the Apenninic orogeny. This manuscript is organized in a logical manner and the data presented appears to be of high quality. Conclusions are for the

most part well supported by the data presented in this study. My main concerns are: 1) A proper assessment of the spatial distribution in outcrop of the different structural diagenetic products is missing. Moreover, their relationship with the anticline and faults is difficult to establish because no orientation... etc. data are provided. It seems like an opportunity was missed to use outcrop exposures to their full extent. 2) Description of cross-cutting relations of the different calcite-filled veins and dolomite cements is vague. Breccias need to be properly described and documented, probably in their own section. Fracture cements described as vein cements (CV) are presumably also present in the host rock, although no description nor documentation are provided. In that case, the use of CV to refer to these cements would be inappropriate. 3) Isotopic signatures and fluid inclusion temperature and salinity ranges of most cement types overlap, but they are used to relate them to different tectonic/fluid events. The one I am having most trouble with is: why would D3 be considered hydrothermal if the temperatures of inclusions in these cements are exactly the same as those of inclusions in D2, which are not considered hydrothermal? Also, Sr/Sr in D3 are much higher than in D4 but both are considered to have been precipitated from the same fluids? 4) The impact of this work would benefit from a discussion of the implications of fault-related dolomitization processes in general, with application for porosity/permeability evolution and fluid-flow in analogous, dolomitized, carbonate-hosted reservoirs and aquifers within similar structures. What is the main driver for fluid circulation? What changes are required in the system to go from dolomite to calcite cementation? When and why did this occur? 5) How is this study better than that of Ronchi (2003)? In which way did it advance the field?

I would recommend publication of this manuscript after the concerns raised here have been properly addressed.

Please refer to the attached PDF for further comments.

Please also note the supplement to this comment:

https://www.solid-earth-discuss.net/se-2018-136/se-2018-136-RC4-supplement.pdf

[Figure]

**Supplement:**

Review: **Fault-controlled dolomitization in the Montagna dei Fiori Anticline (Central Apennines, Italy): Record of a dominantly pre-orogenic fluid migration**

This paper presents field, petrographic, isotopic, and fluid-inclusion thermometric and compositional analyses of the different dolomitization and calcite veining events that affected Lower Jurassic rocks within the Montagna dei Fiori Anticline. The authors conclude that dolomitizing fluids show evidence of interaction with underlying units and therefore infer that dolomitization was fault related and occurred in two main episodes, before and during the Apenninic orogeny.

This manuscript is organized in a logical manner and the data presented appears to be of high quality. Conclusions are for the most part well supported by the data presented in this study. My main concerns are: 1) A proper assessment of the spatial distribution in outcrop of the different structural diagenetic products is missing. Moreover, their relationship with the anticline and faults is difficult to establish because no orientation… etc. data are provided. It seems like an opportunity was missed to use outcrop exposures to their full extent. 2) Description of cross-cutting relations of the different calcite-filled veins and dolomite cements is vague. Breccias need to be properly described and documented, probably in their own section. Fracture cements described as vein cements (CV) are presumably also present in the host rock, although no description nor documentation are provided. In that case, the use of CV to refer to these cements would be inappropriate. 3) Isotopic signatures and fluid inclusion temperature and salinity ranges of most cement types overlap, but they are used to relate them to different tectonic/fluid events. The one I am having most trouble with is: why would D3 be considered hydrothermal if the temperatures of inclusions in these cements are exactly the same as those of inclusions in D2, which are not considered hydrothermal? Also, Sr/Sr in D3 are much higher than in D4 but both are considered to have been precipitated from the same fluids? 4) The impact of this work would benefit from a discussion of the implications of fault-related dolomitization processes in general, with application for porosity/permeability evolution and fluid-flow in analogous, dolomitized, carbonate-hosted reservoirs and aquifers within similar structures. What is the main driver for fluid circulation? What changes are required in the system to go from dolomite to calcite cementation? When and why did this occur? 5) How is this study better than that of Ronchi (2003)? In which way did it advance the field?

This work would benefit from a final edit by a native English speaker. Below I provide some suggestions to improve the language and punctuation, but the authors should take these with care because (warning!) English is not my first language. I would recommend publication of this manuscript after the concerns raised here have been properly addressed.

**Specific comments:**

1) Abstract: The abstract can be shortened substantially, yet it is missing key information. It provides too much detail of some aspects of this work but lacks equivalent detail in other cases. For example, why are calcite-filled veins not mentioned here? Weren't they a main focus of this study? A more succinct and balanced abstract is required. Also, the abstract would benefit from a "punchline" or statement of the broader implications of this work at the end. What did you learn about the extent of dolomitization near faults? How is this relevant for porosity/permeability evolution and fluid flow in dolomitized, carbonate-hosted hydrocarbon reservoirs and aquifers?

2) Introduction: You may want to consider adding a short statement about why some fault-zones become dolomitized but others don't. What are the requirements? What can you learn from outcrops that you cannot from core alone? I would say that the main benefit would be the opportunity to assess the spatial distribution of dolomitized zones, and individual diagenetic events, in 3D. Addition of such a field-relations analysis would greatly improve the impact of this work.

3) Geologic setting: This section is a bit long and could be shortened.

4) Methodology: A few things are missing:
   o How large of a geographic area did you sample?
   o How did you decide which areas to sample for isotopic analyses? Did you image them first? How? How confident are you that you didn't mix different cements when sampling?
   o How many fluid inclusions in your FIAs? What was your reproducibility and error? How did you make sure you did not measure stretched inclusions?
   o Documentation of where hand samples came from is very poor. This can be improved by showing the location of the thin sections on outcrop photos, and their spatial relationship with faults etc. These might need to be included in an appendix due to space restrictions, but it is important.

5) Field observations
   o What is the spatial distribution of the dolomitized geobodies?
   o And of the veined sections?
   o 6 outcrop locations are marked on Figure 2 but distributions of the different types of cements are only shown in one image (figure 5b). These are very important relations to assess fluid pathways and the evolution of dolomitization.
   o What are the orientations of CV1–CV4 cement-bearing fractures?

6) Petrography
   o How much calcite cement is there in the breccias? What are the textures? Why are these not included in your diagenetic evolution analysis? How do cements in breccias relate to those in host rocks? Are cements in the host rocks affected by brecciation? Show examples.

- How did dolomitization affect porosity in both host rock and fault rocks? How does porosity compare between limestones and dolostones?
- Some of the petrographic relationships mentioned need to be backed by images (see my line-specific comments)
- The order in which dolomite cements and vein-calcite cements are mentioned needs to be improved.
- What is the relationship between MC and D1/D2? Where is this documented?
- What is the distribution of CV cements in the host rock (see Laubach, 2003)? This should be properly documented and reported. I don't think vein cement is an appropriate term for these calcite cements. Also, keep in mind that the occlusion of fracture porosity by postkinematic cements can significantly postdate the timing of the opening of the fracture (see Ukar and Laubach, 2016). In other words: the timing of fracturing and cementation are not the same. Keep that in mind in your descriptions.
- The observation of CV3 in breccias is quite interesting. Document and show images. What other cements are there in breccias? How did you establish the relative timing of these cements and others? Breccia cements should not be referred to as vein cements.
- What are the spatial distributions of the different cement types?

7) Geochemistry
- What are the isotopic characteristics of MC and fibrous cements?
- Interpretations, especially for Sr ratios, should be moved to the discussion

8) Fluid inclusions
- Show images of the different types of fluid inclusions, especially the FIAs.
- The graphs used to summarize fluid-inclusion thermometric results are not appropriate because key information is lost. Same for salinity. Please replot the data so that the temperature range for each individual FIA is shown. Did you measure an equal amount of FIAs for each type of cement? Otherwise, frequency would not very meaningful in Fig. 13 because it would be sample and cement availability dependent.
- Why are CV temperatures not shown in these graphs?

9) Discussion
- This section can be significantly shortened by avoiding repetition of results.
- I think parental fluid calculations should be shown in the results section, not in the discussion.
- Use parallel writing style for stable isotopes and Sr discussion.
- D3 shows significantly higher Sr/Sr than D4. How can both be related to the same event and derived from similar fault-related fluids? D1 and D2 are also fault-related. Why the differences in isotopic signatures, especially if all are related to basement-rooted faults???

- o The association of D3 and D4 with bed-parallel and shear fractures is mentioned for the first time in the discussion. This needs to be mentioned in the results. Are the cements themselves sheared? Show evidence.
    - o Discuss the spatial distribution of the different cement and vein/breccia types. What do they indicate about fluid-flow patterns?
    - o The orientation of CV1–CV4-bearing fractures needs to be taken into account in the structural interpretation
    - o Section 5.3: Without a better documentation of the orientations and field relations of the different cements and fracture types it is difficult to assess the validity of the inferred paragenetic sequence and the association of the different cements and structures with tectonic events. Some of the spatial and cross-cutting relationships between different types of cements are first mentioned in this section. Such descriptions should be moved to the results section.
    - o What is the driver for fluid circulation? Why are they Mg-rich fluids? What do fluid-inclusion salinities indicate?
    - o Why do you go from dolomite replacement and cementation to calcite cementation?
    - o How are your findings relevant for porosity/permeability evolution and fluid flow in dolomitized, carbonate-hosted hydrocarbon reservoirs and aquifers associated with similar reservoir-scale structures?
    - o How are your conclusions applicable to dolomitization processes associated with faults in general? How far can dolomitizing fluids travel and to what extent do they alter the mechanics and porosity/permeability of the host rock? What are the consequences for fluid-flow in these rocks?
10) Conclusions
    - o More thought needs to go into the conclusions section.
    - o I don't think enough data are presented in this study, especially of cross-cutting relationships and orientations of the different "deformation structures" to support the structural interpretation presented in the conclusions. For example, where is the evidence that the opening-mode fractures (no orientations or relationships within the anticline are reported!) and normal faults mentioned in this study are associated with contractional tectonics of the Apenninic orogeny?
11) Figures and figure captions need work, especially the model shown in Figure 15 (see comments in figure caption).

Lines 29-31: This needs to be re-written. Layer-parallel shortening would not give place to layer-parallel stylolites. Extensional faulting by itself either. Involvement in the Apenninic thrust wedge of what?

Lines 48-54: May I suggest you take a look at the recently published Ferraro et al. (2019) paper for a description of the diagenetic evolution of carbonate fault rocks in the central and southern Apennines?

Lines 54-58: You may want to consider adding a short statement about why some fault zones become dolomitized but others don't. What are the requirements?

Lines 58-61: What can you learn from outcrops that you cannot from core alone? I would say that the main benefit would be the opportunity to assess the spatial distribution of dolomitized zones, and individual diagenetic events, in 3D. Perhaps a missed opportunity in this work?

Line 67: I don't see Bugarone in Figure 1.

Line 69: I don't see catel Manfrino Dolostones in Figures 1 or 2.

Lines 72-73: How is your study better than Ronchi (2003)?

Lines 76-78: In Figure 2 it appears that dolomitized bodies are found quite far away from faults, beyond the typical lateral extent of fault damage zones. What is the explanation? Why are some faults associates with dolomitization while others aren't? Does it have to do with age of faults? Other factors? This would be a good topic for the discussion.

Line: 84: I would have liked to see more "field mapping" of the extent of D1–D5 and CV1–CV4 in this work.

Line 87: This sentence should start with a different word than "therefore". What provides insights? How?

Lines 88-89. Yes. This needs to be discussed in the discussion.

Lines 90-92: Yes. This needs to be discussed in the discussion.

Line 96: evolution of the Apennines has been proposed to be the result of

Line 98: since the Late Cretaceous

Line 103: The Central Apennines involve

Line 110: lower part of the Burano Formation

Line 115: Deposition of the Hettangian–Sinemurian Calcare Massiccio Formation, with a total thickness …

Line 117: following facies are present

Line 125: deepening-upward trend

Line 137: olistolith model

Lines 138-145: So, does this evidence favor the fault-related model or does this evidence provide an alternative model? Why does this sentence start with 'However"?

Line 151: at a high angle

Line 162-163: 60 samples distributed across how big of an area?

Line 164: What structures?

Line 187: Vienna Pee Dee Belemnite

Line 215: In order to perform high resolution

Lines 220-221: bed-perpendicular stylolites

Lines 224-230: This belongs in the Geological Setting.

Line 225: There is no evidence of dolomitization

Lines 231-234: Location names need to be included in figure captions.

Lines 235-239: This seems out of place. Start by describing mesoscale relations and distributions of dolomitized geobodies. Then focus on hand-sample and petrographic details.

Line 235: in fault cores are typically

Line 237: is "main slip surface" a better term?

Line 238: cut by rather than overprinted. Are dolomite-filled veins intra- or intergranular?

Line 239: calcite cement

Line 241: cross-cutting bedding surfaces

Line 242: from a few meters to hundreds of meters

Line 243: and the lower part of

Line 246: High amplitude (>1 mm), bed-parallel stylolites

Line 247-248: How does porosity differ between limestones and dolostones?

Line 253: grain-supported intervals

Lines 259-260: Evidence?

Line 265: we can't see the displacement mentioned in Figure 2A, site 1

Line 267-268: Are they overprinting or overgrowing? Show evidence. We also cannot see the distribution of the different cements at outcrop scale.

271-272: On what basis did you establish that the replacive dolomite within the host rock (D1) and lining fractures is the same?

Lines 275-276: solid inclusions of what? Insert figure call out for concentric zonation.

Line 281: sweeping extinction

Line 282: In some crystals, one… what types of solid and fluid inclusions?

Lines 286-291: Mark locations on map/figure captions and call out the figure.

Line 290-291: Scaglia Formation in the hanging wall.

Lines 293-295: This needs to be moved up

Line 305: bed-parallel shear fractures

Lines 308 on: There is a problem with CV introduction. What does it stand for? Must introduce them in chronological order. If the calcite cement is in veins it is most likely in the host rock as well (see Laubach, 2003). I don't think calling it vein cement is appropriate.

Line 308: What porosity?

Lines 318-320: Show outcrop photo?

Line 319: bed-perpendicular rather than bedding because that's what you use elsewhere. Make sure term usage is consistent throughout.

Line 321: bed-parallel stylolites. CV1 usually shows (often means time. Correct elsewhere in the manuscript).

Lines 324-326: Show image of CV2 in tension gashes

Line 332: extensional fault's master (main?) plane

Line 339-340: More evidence that the use of CV to refer to calcite cements that occur in a variety of textures and petrographic relations is not appropriate.

Lines 361-364: Why is this mentioned here and not with the rest of the calcite cements?

Line 396: I see 3 values plotted for Scaglia.

Lines 398-399: Interpretation. Move to discussion. Same comment for previous paragraphs.

Lines 401-411: This belongs in the Methods section.

Lines 409 and 439: all-liquid inclusions

Lines 439-445: This belongs in the Methods section.

Lines 469-470: This belongs in the methods/results. Why did you avoid them? Could mottled D be a different type than those reported?

Lines 471-474: Move to methods. Report parental fluid calculations in the results section.

Line 478: Progressively higher than what?

Line 481: siliciclastic rocks,… Correct here and elsewhere.

Line 493: , or values recorded… Add references.

Lines 499 and 505: Replace comparable with similar. Here and elsewhere.

Line 507: stylolitization of the host rock (otherwise we do not know what dissolution etc. you are referring to).

Lines 527-528: fluids related to Late Messinian… overlying Upper Miocene Laga Formation and their possible…

Line 543: burial-related temperature

Line 544: it is unlikely that the..

Line 546: located at higher stratigraphic levels

Line 564: calcite cements (FC) in grain-supported stratigraphic levels of the CMF is interpreted to be…

Line 570: bed-parallel stylolties

Line 574 and 575: are cut by

Line 579: Figure 15A call out.

Line 585: We cannot see the distribution described in Figure 2A.

Line 587: attributed to post-rift

Line 588: Although an absolute age cannot be provided,

Line 603: bed-parallel fractures. This is the first mention of shear veins for D4. Are the cements sheared? Show evidence.

Line 604: Contractional deformations? How? Describe relationships better. Bed-perpendicular dilation alone would not cause shear.

Lines 608-610: First mention of this. Move to results.

Lines 618-619: bed-parallel veins

Line 622: fragments suggests that… late-stage evolution

Line 624: bed-perpendicular stylolites

Line 624-629. This is way too long. In any case, there is new information here that needs to be moved to the results section.

Line 629: low homogenization temperatures of fluid inclusions trapped within these cements

Lines 638-642: So, how is your study better? How are your conclusions applicable to dolomitization processes associated with faults in general?

Figures: Documentation of where samples came from is poor. Locations of samples need to be shown on outcrop photos and/or detailed maps. Add in appendix if space is limited.

Figure 1: Tiny name in a) is unreadable. Mark location of cross-section (A-A') shown in d).

Figure 2: Add names of each field site to the figure caption.

Figure 3: The picture in a) is too close up to see the context. Mark distribution of D1 etc. as in Fig 5d. Why isn't there more on these breccias (c) in the manuscript? Explain what arrows point to. Pressure solution seams. You are not showing intensity (it would be a number). Perhaps say showing abundant pressure solution seams. What are the abutting relationships? Which abuts which? Move arrow in b) so that the vein is actually visible. Are the other white pods also considered "veins"?

Figure 4: Show spatial distribution of D1, D3, CV1… etc. at the outcrop scale. The sentence seems to say that CV1 veins are dolomitized. Is that what you really mean? What is the dolomite type in b) and c)? Good opportunity to show cross-cutting relationships summarized in Figure 14.

Figure 5: What field site(s) and formation(s) are these from? In b) the zone shown in c) is marked as only having D1 but as D1 + D2 in c). Which one is correct? Any CV2-CV4 here?

Figure 6: What field site is this? rimmed by fibrous cements (FC), which are overgrown by mosaic cements (MC). overprinted. bed-parallel stylolites. D1cements lining a fracture. What do

arrows point to in f) If it is D1, then what is to the right of it? Line 1123: which is cemented by CV1 in the center.

Figure 7: What field site(s) and formation(s) are these from? What is beyond D3 in e) and f)? Is D3 only present in breccias in this site? What is the CL signal of D3 in breccias and how did you establish correspondence with D3 in host rocks? What is the context of the sample? in e) and f)

Figure 8: What field site(s) and formation(s)? D3 and D4 are not cross-cutting but it appears that D4 overgrows D3. What is the D4 arrow in b) exactly pointing to? What cement is in the rhomb on the upper right corner? d) I am having a hard time seeing the microfracture. What CV is this? Or do you have dolomite-filled veins as well? Where are these described? What other cements are in e) and f) and what do arrows point to? Lines 1141-1150 belongs in the discussion.

Figure 9: What field site(s) and formation(s)? If D4 also occurs in fractures why is it not called vein cement as in your CV scheme? What other cements and/or host rock are in these photographs? Add labels.

Figure 10: These photographs are too close up to see the context. Outlining obscures fractures in a). c-d) These do not look like tension gashes to me (as mentioned in text?). Mislabeled as CV2 on the picture (?) but CV3 in the figure caption.

Figure 11: What field site(s) and formation(s)? What is a) a sample of? And the rest? Show field context.

Figure 12: Why aren't these figures in color? The symbols are too similar and they are hard to distinguish from one another. I would assign a color to each formation and a symbol to each diagenetic feature.

Figure 13: These plots are not very useful because key information is lost. Plot homogeneization and ice melting T ranges so that variability within individual FIAs is captured. Color would help, Also, where are the data for CV cements?

Figure 15: The fracture in a) would not have that orientation if sigma 1 were vertical. Indicate which fault you are referring to in b). The vein in b) would not develop in that orientation if sigma 1 were horizontal. Also, keep in mind that the timing of cementation of the veins by postkinematic cements (see Ukar and Laubach, 2016) postdates timing of opening of the fracture. Don't mix the two! I had no idea that CV4 is restricted to the MdFF until now, because it is not mentioned anywhere in the text. How do you reconcile D3 to be surrounding breccia clasts within the MdFF in this model and sequence?

Estibalitz Ukar
Research Associate
Jackson School of Geosciences
The University of Texas at Austin

---

## Author Comment (AC4) · 25 Apr 2019

Dear Dr. Ukar, Thank you for careful and detailed review of the manuscript. We greatly appreciate your constructive comments. They greatly helped us to improve the quality of the manuscript. We have tried our best to address your suggestions in this new version of the manuscript. Regarding the advancement of our research in comparison with the work previously presented by Ronchi et al. (2003), the current study gives much more details about the dolomite characterization and their relation to the structural evolution of the anticline on the regional and local scale. Furthermore,

the obtained geochemical and microthermometry analyses do not confirm the role of marly or shaly basinal successions in providing the Mg-rich fluids during the first event of dolomitization (i.e. syn-rift), as proposed by Ronchi et al. (2003). We have tried to be modest in criticizing the latter authors limited research since the current research was build up on their findings. Another important question about the studied dolomites was the role of Scaglia Formation in providing the Mg-rich fluids during compression, because this formation is juxtaposed with the dolostones by the Montagna dei Fiori Fault. Our results do not support this hypothesis. Moreover, we show that the dolomitization predates the observed juxtaposition. During our research, we also performed some other advance analyses such as clumped isotopes and U/Pb dating. However, the consecutive overgrowth pattern of dolomites and difficulties in isolating them to get enough and good quality samples increased the uncertainty in the results. Therefore, we decided not to include those data in the manuscript. In the Montagna dei Fiori Anticline, the structures and their relative chronology are very complicated. A comprehensive structural study on the evolution of the Montagna dei Fiori Anticline was performed parallel with the current study, and published by Storti et al. (2018) in Tectonics. The target of the current study was to focus on dolomitization, and to use the structural model proposed by Storti et al. (2018) to deduce the most likely timing for dolomitization. The distribution of dolomitization and sampled locations are way larger than the out crop photo scale. The dolomitized intervals are tens of meters mostly exposed in vertical to subvertical outcrops. To be able to show their 3D distribution properly a photogrammetry or LiDAR imaging is required. The brecciated zones are mostly clast-support with minor calcite and negligible dolomite cement. Moreover, a detailed classification of breccia is not the focus of this research and does not give relevant information regarding this dolomitization case study. This new version of the manuscript has been reviewed by a native English speaker. Best regards, Mozafari et al.

Please also note the supplement to this comment:

[Figure]

https://www.solid-earth-discuss.net/se-2018-136/se-2018-136-AC4-supplement.zip

---

## Author Comment (AC5) · 30 Apr 2019

Dear Dr. Hendry, Thank you for the references and your constructive suggestions. Mozafari et al.
* * *

---

## Author Response (AR1)

**Review of "Fault-related dolomitization in the Montagna dei Fiori Anticline (Central Apennines, Italy): Record of a dominantly pre-orogenic fluid migration. Se-2018-136**

This manuscript presents field, petrographic and geochemical data from non-stratabound dolomites in a complex tectonic setting, and interprets their geofluid origin (parent fluids, timing) in the context of the tectonostratigraphic history.

It is coherent and logically organised. Aspects of the written English need minor improvement (punctuation, plurals, word order, etc); it will benefit from a final revision by a native English speaker.

The data are generally of good quality, and the interpretations are mostly justified from the results presented. In any study such as this, with limitations imposed by the ability to sample all the phases, there is necessarily some latitude or flexibility in the deductions that can be made. However, the authors do a good job of considering alternative possibilities for the fluid sources and timings.

My only issue with the paper is that the authors have not really considered whether there are wider implications or generic advances that can be made from the research. It presents itself as a case study, albeit one with a good integration of structural and diagenetic data. But what is the wider impact that will attract a non-specialist readership? Within the paper the authors all but admit that their findings are only modestly advanced from those of Ronchi and co-workers fifteen years ago (lines 638-642). I had hoped to see more progression in the science, and maybe the authors need to more thoroughly and critically evaluate the Ronchi model in the light of their new data. They could also work the structural data more – rather than just considering fault orientations and timings, what about the character and extent of the damage zones associated with different fault types / generations, and their relationship to the size and shape of dolomite bodies? What determines the lateral extent of dolomites? Is it other faults / fractures, or a gradual reaction front?

One generic aspect that the authors could address is implications for reservoir potential in analogues for this setting. The preponderance of planar dolomite is significant because planar dolomite is usually very beneficial for porperm (unlike many examples of hydrothermal dolomitization that feature tight nonplanar dolomites). Are there dolomitised plays in the Middle East that this study could be compared to (Zagros Mountains for example?), or maybe in Mexico?

Another factor of interest, largely by-passed in the text, is what drove the fluid circulation necessary to cause massive dolomitization when the low temperatures argue against a hydrothermal syn-rift system. Can the authors attempt a mass balance to estimate the order of magnitude fluid volumes needed? Maybe the dolomitization occurred in the down-flowing (cool) limb of a convection cell established on syn-rift faults that breached contemporary sea bed? Or does the dolomite zoning imply a pulsed fluid flow associated with strain cycling or seismic valving? I recommend the recent papers by Hollis and others on the Hammam Faraun fault and related syn-rift dolomitization. Are the D1-2 dolomites formed in a similar manner to this geologically younger example? Likewise, with the later dolomitization, which structures would have been open during compressional tectonics and able to serve as conduits for substantial fluid volumes?

If the authors address these issues their paper, which is already technically good, it will have much greater impact and interest across the sedimentary and structural geoscience community.

I have some more **specific comments** – these are tagged by line number or Figure number:

Line 43: The paper describes the role of evaporite-sourced fluids in the dolomitization process, but I am not sure that it illustrates a controlling role of evaporitic detachments. These may have influenced the tectonic development, but if it is believed that they directly controlled the dolomitization this needs to be specifically discussed later in the paper. Emphasized more in the text

Note the abstract is quite long-winded. It would be good to make it more succinct and punchier. Addressed

Line 69: The Castel Manfrino Dolostones are not labelled in Fig. 1 or Fig. 2b. Added in Fig.1

Line 73, 76: Did Ronchi (2003) base her study on the mapping of Mattei (1987)? Maybe there needs to be a couple of sentences describing Ronchi's findings so that it can be more clearly shown that the understanding has moved on. Added to text

Line 79-83: This is very long-winded and vague. It either needs to be shortened or to include specific details. The sentence is shortened.

Line 129: Given that the early dolomitization (D1-2) is later ascribed to the syn-rift stage, it would be useful to briefly describe the facies and architectural character of the syn-rift carbonates. For example, were they preferentially developed on footwall highs, in which case there was likely a juxtaposition of permeable high energy facies against faults that later hosted fluid flow? Addressed

Line 135-137: It may be a matter of debate, but the authors need to either provide the conflicting evidence or at least express a view and justify it. The sentence is deleted as further discussion is not the focus of this research

Line 152: Can the Montagna dei Fiori fault be indicated / labelled on Fig. 1d? Addressed

Line 164: Ground truthing implies that the features had previously been mapped out using remote data. If so, this should be included in the methods. Addressed

Line 167: So far as I can see, the Sibley and Gregg (1987) terminology (planar-e, planar-s, nonplanar) is <not> used anywhere in the paper, so either it needs to be incorporated or this sentence should be removed. The sentence and related references are removed

Line 186: Can the reproducibility (±0.1‰) be smaller than the precision (±0.2‰)? The inter-lab reproducability is ±0.1‰ means the measurement for the same sample in two different labs has a difference of ±0.1‰

Line 224: Bugarone Formation is not shown on Fig. 1 or Fig 2. Addressed
Line 231: Is the wider distribution of dolomitized intervals related to the topography of the valley and the exposures? If not, what is the relationship? Addressed

Line 238: I suggest not using "overprinted", which implies the original fabric / lithology is lost. Why not just use "cross-cut"? (or even just "cut") Addressed

Line 258 and elsewhere: "Dull" is not a colour! Addressed

Line 261 onwards: There could be a bit more detail on the dolomite distribution and fabric with respect to host rock facies. Is it all texturally destructive, is there any textural or mineralogical selectivity, were grainy or muddy facies preferentially dolomitized (controls by permeability versus reactive surface area…….?) Addressed

Line 272: There is an issue because CV1, CV2 etc. are introduced before they have been defined and described. I suggest starting section 4.2 with a paragenetic summary to alleviate this problem. Addressed, paragenesis is presented in Fig. 14.

Line 278: By using "frequently" the text suggests that sometimes (infrequently) D2 post-dates bed-parallel stylolites. Is that the case? Addressed

Line 284-285: This sentence needs a figure citation. Addressed

Line 297: Repetition of "euhedral to anhedral" (cf. line 295) – note this is not Sibley and Gregg terminology. Nor is "tightly packed texture" in line 279. Addressed

Line 305: I do not think one dolomite can "recrystallize" another. Recrystallization is a solid-state process that increased mineral stability. To demonstrate it might need data on the ordering, crystallinity and stoichiometry of D1/2 versus D4 (do the authors have any XRD data?). What is more likely is that D4 has locally replaced D1 and D2 by a dissolution-precipitation mechanism. However, the text lacks a clear description of the evidence for replacement. I recall papers by Mazzullo and by Machel that discuss this – it would be good to list the criteria for this case. Addressed

Line 330: In Fig. 11C, D the dolomite does not appear to be yellow-orange, it looks more like orange-brown. Addressed

Line 332: How wide was the extensional fault master plane? Please supply the range of widths (and lengths where possible) of the different vein generations. It is addressed for vein generations.

Line 335-336: What is meant by "with no evidence of physical disruption"? Does it mean that CV3 always passively overgrows D5 in voids and never cuts it? If so, it is easier to say this. Addessed

Line 336: Translucent is not a colour. Addressed

Line 334: What colours are the zones? Addressed

Line 367-368: Rather than "the presumable Lower Jurassic marine dolomites" – which are hypothetical – it is better to say the values are lower than those expected for Lower Jurassic seawater dolomites. Addressed

Line 391: "While….." indicates there should be a second part to the sentence. Addressed

Line 396: What is the lithology of these samples? Addressed

Line 401: Please add a sentence or two on the fluid inclusion petrography and distribution – do inclusions follow growth zones or are they randomly distributed? Are they all primary or are some pseudosecondary? What are their shapes and what are the liquid:vapour ratios in the 2-phase examples? Also, in reporting the results for different cements please give the number of inclusions the ranges are based on (n=). Addressed

Line 409-410: What is the purpose of nucleating a bubble for measurement of freezing temperatures? Addressed

Line 477-478: This sentence appears out of place or at least needs clarification. More than two values are needed to demonstrate a progression. Addressed

Line 503-507: Yes, this makes sense if the veins are filling tension gashes associated with stylolites – such as system is likely to be buffered by the dissolving carbonate. Maybe make this point more explicitly, and contrast with vein types that were more extensive and would have allowed allochthonous fluids to pass through with minimal host-rock interaction. Addressed

Line 518: Hendry et al. (2015) did not discuss $^{13}C$ enrichment from $CO_2$ outgassing due to evaporation. They made the point that negative covariance in C and O isotopes within veins could be due to precipitation during $CO_2$ outgassing related to pressure changes. Addressed

Line 524-551: This is good but is a very long paragraph. Can it be made more succinct? Addressed

Line 550-551: The final sentence needs rewriting; what was confined, the thrust wedge or the fluids? Addressed

Line 556: In the preceding section there is very little mention of fluid mixing. Could the poorly correlated Th, salinity and stable isotopic data reflect precipitation from allochthonous fluids as they mixed with in situ fluids (and cooled)? Degrees of mixing (and of water-rock interaction) may have different from fault to fault – is it really likely that the hydrogeological systems was as simple as is being presented here? The obtained data show no systematic variations from fault to fault. Thus, existance of different local hydrological systems cannot be addressed. The sentence is revised ad completed

Line 575-579: Please rewrite this sentence – it tries to say too many things at the same time. Addressed

Line 584: Doesn't the displacement of D1-2 on these faults indicate that the dolomite formed before faulting? What is the critical evidence that it is genuinely syn-rift? The syn-rift deposits (Corniola Formation) is affected by these dolomites. Addressed more in the text

Line 590: if D1-2 were related to basement-cutting faults, why are the Sr isotope values much less elevated than for D3 and D4? There is no other alternative for radiogenic Sr source. This is the case for all of the dolomite types. Maybe less basement deriven fluids were involved in D1.

Line 656-657: Please explain how hydrothermal fluids were able to circulate in the compressional tectonic regime – which structures were able to be in tension and therefore transmissive rather than sealing? Addressed

Line 1139: The cross-cutting relationship in Fig. 8a, b is not very evident. Addressed

Lines 1142-1148: Should this discussion be in the main text rather than in the caption? This has been made to avoid a longer discussion.

Fig. 2b: It would help if the colours and ornaments matched Fig. 8a (e.g. Salinello Fm). The text size needs to be increased for better legibility. Addressed

The stereonet data are good, but very little use is made of them in the text of the paper. Addressed in the figure caption

Fig. 5: Please increase the text size and make 5c larger – it is too small to see clearly. Addressed

Figs 6-9, 11: Some of the CL images could be a bit sharper and maybe with increased contrast to better discriminate the dolomite types. Addressed

Fig. 12: The symbols for D3 vs. CV3 and mixed dolomite vs. CV1 are too similar (especially given the small size). I am also not clear how Fig. 12b relates to Fig. 12a; maybe split the legend between the two plots according to what is in them – that might help. Addressed

Fig. 14: How were the burial temperatures in the burial history determined? What assumptions are they based on? Addressed

Fig. 15: I like this figure but I'm still not sure what the fluid flow pathway is in (b). Maybe a broader tectonic context diagram is needed showing expulsion of fluids from the foreland (if that is where they are coming from?). The fluids were migrated from hinterland (now indicated on the sketch) rather than forland.

I hope these comments are helpful, and I look forward to seeing the paper published in revised form.

Jim Hendry

Tullow Oil Ltd, Dublin

**Specific comments:**

1) Abstract: The abstract can be shortened substantially, yet it is missing key information. It provides too much detail of some aspects of this work but lacks equivalent detail in other cases. For example, why are calcite-filled veins not mentioned here? Weren't they a main focus of this study? A more succinct and balanced abstract is required. Also, the abstract would benefit from a "punchline" or statement of the broader implications of this work at the end. What did you learn about the extent of dolomitization near faults? How is this relevant for porosity/permeability evolution and fluid flow in dolomitized, carbonate-hosted hydrocarbon reservoirs and aquifers? Addressed

2) Introduction: You may want to consider adding a short statement about why some fault-zones become dolomitized but others don't. What are the requirements? What can you learn from outcrops that you cannot from core alone? I would say that the main benefit would be the opportunity to assess the spatial distribution of dolomitized zones, and individual diagenetic events, in 3D. Addition of such a field-relations analysis would greatly improve the impact of this work. Addressed

3) Geologic setting: This section is a bit long and could be shortened. Addressed

4) Methodology: A few things are missing:
    o How large of a geographic area did you sample? Addressed
    o How did you decide which areas to sample for isotopic analyses? Did you image them first? How? How confident are you that you didn't mix different cements when sampling? Addressed
    o How many fluid inclusions in your FIAs? What was your reproducibility and error? How did you make sure you did not measure stretched inclusions? Addressed

    o Documentation of where hand samples came from is very poor. This can be improved by showing the location of the thin sections on outcrop photos, and their spatial relationship with faults etc. These might need to be included in an appendix due to space restrictions, but it is important. Explained in author response

5) Field observations
    o What is the spatial distribution of the dolomitized geobodies? Addressed
    o And of the veined sections?
    o 6 outcrop locations are marked on Figure 2 but distributions of the different types of cements are only shown in one image (figure 5b). These are very important relations to assess fluid pathways and the evolution of dolomitization. Addressed
    o What are the orientations of CV1–CV4 cement-bearing fractures? Shown in Storti et al. (2018)

6) Petrography
    o How much calcite cement is there in the breccias? What are the textures? Why are these not included in your diagenetic evolution analysis? How do cements in breccias relate to those in host rocks? Are cements in the host rocks affected by brecciation? Show examples. Addressed

- How did dolomitization affect porosity in both host rock and fault rocks? How does porosity compare between limestones and dolostones? Addressed
- Some of the petrographic relationships mentioned need to be backed by images (see my line-specific comments) Addressed
- The order in which dolomite cements and vein-calcite cements are mentioned needs to be improved.
- What is the relationship between MC and D1/D2? Where is this documented? Addressed
- What is the distribution of CV cements in the host rock (see Laubach, 2003)? This should be properly documented and reported. I don't think vein cement is an appropriate term for these calcite cements. Also, keep in mind that the occlusion of fracture porosity by postkinematic cements can significantly postdate the timing of the opening of the fracture (see Ukar and Laubach, 2016). In other words: the timing of fracturing and cementation are not the same. Keep that in mind in your descriptions. Addressed
- The observation of CV3 in breccias is quite interesting. Document and show images. What other cements are there in breccias? How did you establish the relative timing of these cements and others? Breccia cements should not be referred to as vein cements. Explained in author response
- What are the spatial distributions of the different cement types? Addressed

7) Geochemistry
- What are the isotopic characteristics of MC and fibrous cements? Addressed
- Interpretations, especially for Sr ratios, should be moved to the discussion.

8) Fluid inclusions
- Show images of the different types of fluid inclusions, especially the FIAs.
- The graphs used to summarize fluid-inclusion thermometric results are not appropriate because key information is lost. Same for salinity. Please replot the data so that the temperature range for each individual FIA is shown. Did you measure an equal amount of FIAs for each type of cement? Otherwise, frequency would not very meaningful in Fig. 13 because it would be sample and cement availability dependent.
- Why are CV temperatures not shown in these graphs? For a higher focus on dolomitization case study

9) Discussion
- This section can be significantly shortened by avoiding repetition of results.
- I think parental fluid calculations should be shown in the results section, not in the discussion. To emphasize on the nature of parental fluids, we prefer to keep SMOW values in discussion section

- Use parallel writing style for stable isotopes and Sr discussion.???
- D3 shows significantly higher Sr/Sr than D4. How can both be related to the same event and derived from similar fault-related fluids? D1 and D2 are also fault-related. Why the differences in isotopic signatures, especially if all are related to basement-rooted faults??? Addressed

- The association of D3 and D4 with bed-parallel and shear fractures is mentioned for the first time in the discussion. This needs to be mentioned in the results. Are the cements themselves sheared? Show evidence. Addressed in petrography

- Discuss the spatial distribution of the different cement and vein/breccia types. What do they indicate about fluid-flow patterns? The breccia types in MDF are not diverse. More details on bereccia is out of focus of this study.

- The orientation of CV1–CV4-bearing fractures needs to be taken into account in the structural interpretation. A detailed structural interpretation is discussed in Storti et al. 2018. Its now more emphasized in this manuscript.

- Section 5.3: Without a better documentation of the orientations and field relations of the different cements and fracture types it is difficult to assess the validity of the inferred paragenetic sequence and the association of the different cements and structures with tectonic events. Some of the spatial and cross-cutting relationships between different types of cements are first mentioned in this section. Such descriptions should be moved to the results section.

- What is the driver for fluid circulation? Why are they Mg-rich fluids? What do fluid-inclusion salinities indicate? Addressed in section 5.2

- Why do you go from dolomite replacement and cementation to calcit cementation? Addressed

- How are your findings relevant for porosity/permeability evolution and fluid flow in dolomitized, carbonate-hosted hydrocarbon reservoirs and aquifers associated with similar reservoir-scale structures? Addressed

- How are your conclusions applicable to dolomitization processes associated with faults in general? How far can dolomitizing fluids travel and to what extent do they alter the mechanics and porosity/permeability of the host rock? What are the consequences for fluid-flow in these rocks? Addressed

10)

Conclusions

- More thought needs to go into the conclusions section.

- I don't think enough data are presented in this study, especially of cross-cutting relationships and orientations of the different "deformation structures" to support the structural interpretation presented in the conclusions. For example, where is the evidence that the opening-mode fractures (no orientations or relationships within the anticline are reported!) and normal faults mentioned in this study are associated with contractional tectonics of the Apenninic orogeny? All are discussed in details in Storti et al. 2018.

11) Figures and figure captions need work, especially the model shown in Figure 15 (see comments in figure caption).

Lines 29-31: This needs to be re-written. Layer-parallel shortening would not give place to layer-parallel stylolites. Extensional faulting by itself either. Involvement in the Apenninic thrust wedge of what? Addressed

Lines 48-54: May I suggest you take a look at the recently published Ferraro et al. (2019) paper for a description of the diagenetic evolution of carbonate fault rocks in the central and southern Apennines? Addressed

Lines 54-58: You may want to consider adding a short statement about why some fault zones become dolomitized but others don't. What are the requirements? Addressed

Lines 58-61: What can you learn from outcrops that you cannot from core alone? I would say that the main benefit would be the opportunity to assess the spatial distribution of dolomitized zones, and individual diagenetic events, in 3D. Perhaps a missed opportunity in this work?

Line 67: I don't see Bugarone in Figure 1. Addressed

Line 69: I don't see catel Manfrino Dolostones in Figures 1 or 2. All the dolomitized intervals are called

Castel Manfrino Dolostones

Lines 72-73: How is your study better than Ronchi (2003)? Addressed

Lines 76-78: In Figure 2 it appears that dolomitized bodies are found quite far away from faults, beyond the typical lateral extent of fault damage zones. What is the explanation? Why are some faults associates with dolomitization while others aren't? Does it have to do with age of faults? Other factors? This would be a good topic for the discussion.

Line: 84: I would have liked to see more "field mapping" of the extent of D1–D5 and CV1–CV4 in this work.

Line 87: This sentence should start with a different word than "therefore". What provides insights? How? Addressed

Lines 88-89. Yes. This needs to be discussed in the discussion. Addressed

Lines 90-92: Yes. This needs to be discussed in the discussion. Addressed

Line 96: evolution of the Apennines has been proposed to be the result of Addressed

Line 98: since the Late Cretaceous Addressed

Line 103: The Central Apennines involve OK

Line 110: lower part of the Burano Formation Addressed

Line 115: Deposition of the Hettangian–Sinemurian Calcare Massiccio Formation, with a total thickness … Addressed

Line 117: following facies are present Addressed

Line 125: deepening-upward trend Addressed

Line 137: olistolith model This sentence is deleted

Lines 138-145: So, does this evidence favor the fault-related model or does this evidence provide an alternative model? Why does this sentence start with 'However"? This sentence is deleted

Line 151: at a high angle Addressed

Line 162-163: 60 samples distributed across how big of an area? Addressed

Line 164: What structures? Addressed

Line 187: Vienna Pee Dee Belemnite Addressed

Line 215: In order to perform high resolution Addressed

Lines 220-221: bed-perpendicular stylolites Addressed

Lines 224-230: This belongs in the Geological Setting. Addressed

Line 225: There is no evidence of dolomitization Addressed

Lines 231-234: Location names need to be included in figure captions. Addressed

Lines 235-239: This seems out of place. Start by describing mesoscale relations and distributions of dolomitized geobodies. Then focus on hand-sample and petrographic details. Addressed

Line 235: in fault cores are typically Addressed

Line 237: is "main slip surface" a better term? Addressed

Line 238: cut by rather than overprinted. Are dolomite-filled veins intra- or intergranular?

Line 239: calcite cement Addressed

Line 241: cross-cutting bedding surfaces Addressed

Line 242: from a few meters to hundreds of meters Addressed

Line 243: and the lower part of Addressed

Line 246: High amplitude (>1 mm), bed-parallel stylolites Addressed

Line 247-248: How does porosity differ between limestones and dolostones?

Line 253: grain-supported intervals Addressed

Lines 259-260: Evidence? Addressed

Line 265: we can't see the displacement mentioned in Figure 2A, site 1

Line 267-268: Are they overprinting or overgrowing? Show evidence. We also cannot see the distribution of the different cements at outcrop scale.

271-272: On what basis did you establish that the replacive dolomite within the host rock (D1) and lining fractures is the same?

Lines 275-276: solid inclusions of what? Insert figure call out for concentric zonation. Addressed

Line 281: sweeping extinction Addressed

Line 282: In some crystals, one… what types of solid and fluid inclusions? Addressed

Lines 286-291: Mark locations on map/figure captions and call out the figure. Addressed

Line 290-291: Scaglia Formation in the hanging wall. Addressed

Lines 293-295: This needs to be moved up Addressed

Line 305: bed-parallel shear fractures Addressed

Lines 308 on: There is a problem with CV introduction. What does it stand for? Must introduce them in chronological order. If the calcite cement is in veins it is most likely in the host rock as well (see Laubach, 2003). I don't think calling it vein cement is appropriate. Addressed

Line 308: What porosity? Addressed

Lines 318-320: Show outcrop photo?

Line 319: bed-perpendicular rather than bedding because that's what you use elsewhere. Make sure term usage is consistent throughout. Addressed

Line 321: bed-parallel stylolites. CV1 usually shows (often means time. Correct elsewhere in the manuscript). Addressed

Lines 324-326: Show image of CV2 in tension gashes Addressed

Line 332: extensional fault's master (main?) plane Addressed

Line 339-340: More evidence that the use of CV to refer to calcite cements that occur in a variety of textures and petrographic relations is not appropriate.

Lines 361-364: Why is this mentioned here and not with the rest of the calcite cements? Its ordered based on relative timig

Line 396: I see 3 values plotted for Scaglia. Addressed

Lines 398-399: Interpretation. Move to discussion. Same comment for previous paragraphs.Addressed

Lines 401-411: This belongs in the Methods section. Addressed

Lines 409 and 439: all-liquid inclusions Addressed

Lines 439-445: This belongs in the Methods section. Addressed

Lines 469-470: This belongs in the methods/results. Why did you avoid them? Could mottled D be a different type than those reported? Addressed

Lines 471-474: Move to methods. Report parental fluid calculations in the results section.

Line 478: Progressively higher than what? Fixed

Line 481: siliciclastic rocks,… Correct here and elsewhere. Addressed

Line 493: , or values recorded… Add references. Addressed

Lines 499 and 505: Replace comparable with similar. Here and elsewhere. Addressed

Line 507: stylolitization of the host rock (otherwise we do not know what dissolution etc. you are referring to). Addressed

Lines 527-528: fluids related to Late Messinian… overlying Upper Miocene Laga Formation and their possible…Addressed

Line 543: burial-related temperature Addressed

  Line 544: it is unlikely that the.. Addressed

Line 546: located at higher stratigraphic levels Addressed

Line 564: calcite cements (FC) in grain-supported stratigraphic levels of the CMF is interpreted to be… Addressed

  Line 570: bed-parallel stylolties Addressed

  Line 574 and 575: are cut by Addressed

  Line 579: Figure 15A call out. Addressed

Line 585: We cannot see the distribution described in Figure 2A.

Line 587: attributed to post-rift Addressed

Line 588: Although an absolute age cannot be provided, Addressed

Line 603: bed-parallel fractures. This is the first mention of shear veins for D4. Are the cements sheared? Show evidence. Cements are not sheared. Addressed

Line 604: Contractional deformations? How? Describe relationships better. Bed-perpendicular dilation alone would not cause shear.

Lines 608-610: First mention of this. Move to results.

Lines 618-619: bed-parallel veins Addressed

Line 622: fragments suggests that… late-stage evolution Addressed

Line 624: bed-perpendicular stylolites Addressed

Line 624-629. This is way too long. In any case, there is new information here that needs to be moved to the results section. Addressed

Line 629: low homogenization temperatures of fluid inclusions trapped within these cements Addressed

Lines 638-642: So, how is your study better? How are your conclusions applicable to dolomitization processes associated with faults in general?

Figures: Documentation of where samples came from is poor. Locations of samples need to be shown on outcrop photos and/or detailed maps. Add in appendix if space is limited.

Figure 1: Tiny name in a) is unreadable. Mark location of cross-section (A-A') shown in d).

Figure 2: Add names of each field site to the figure caption. Addressed

Figure 3: The picture in a) is too close up to see the context. Mark distribution of D1 etc. as in Fig 5d. Why isn't there more on these breccias (c) in the manuscript? Explain what arrows point to. Pressure solution seams. You are not showing intensity (it would be a number). Perhaps say showing abundant pressure solution seams. What are the abutting relationships? Which abuts which? Move arrow in b) so that the vein is actually visible. Are the other white pods also considered "veins"?

Figure 4: Show spatial distribution of D1, D3, CV1… etc. at the outcrop scale. The sentence seems to say that CV1 veins are dolomitized. Is that what you really mean? What is the dolomite type in b) and c)? Good opportunity to show cross-cutting relationships summarized in Figure 14.

Figure 5: What field site(s) and formation(s) are these from? In b) the zone shown in c) is marked as only having D1 but as D1 + D2 in c). Which one is correct? Any CV2-CV4 here? Addressed

Figure 6: What field site is this? rimmed by fibrous cements (FC), which are overgrown by mosaic cements (MC). overprinted. bed-parallel stylolites. D1cements lining a fracture. What do arrows point to in f) If it is D1, then what is to the right of it? Line 1123: which is cemented by CV1 in the center. Addressed

Figure 7: What field site(s) and formation(s) are these from? What is beyond D3 in e) and f)? Is D3 only present in breccias in this site? What is the CL signal of D3 in breccias and how did you establish correspondence with D3 in host rocks? What is the context of the sample? in e) and f) Addressed

Figure 8: What field site(s) and formation(s)? D3 and D4 are not cross-cutting but it appears that D4 overgrows D3. What is the D4 arrow in b) exactly pointing to? What cement is in the rhomb on the upper right corner? d) I am having a hard time seeing the microfracture. What CV is this? Or do you have dolomite-filled veins as well? Where are these described? What other cements are in e) and f) and what do arrows point to? Lines 1141-1150 belongs in the discussion. Addressed

Figure 9: What field site(s) and formation(s)? If D4 also occurs in fractures why is it not called vein cement as in your CV scheme? What other cements and/or host rock are in these photographs? Add labels. Addressed

Figure 10: These photographs are too close up to see the context. Outlining obscures fractures in a). c-d) These do not look like tension gashes to me (as mentioned in text?). Mislabeled as CV2 on the picture (?) but CV3 in the figure caption. Addressed

Figure 11: What field site(s) and formation(s)? What is a) a sample of? And the rest? Show field context. Addressed

Figure 12: Why aren't these figures in color? The symbols are too similar and they are hard to distinguish from one another. I would assign a color to each formation and a symbol to each diagenetic feature. Addressed

Figure 13: These plots are not very useful because key information is lost. Plot homogeneization and ice melting T ranges so that variability within individual FIAs is captured. Color would help, Also, where are the data for CV cements?

Figure 15: The fracture in a) would not have that orientation if sigma 1 were vertical. Indicate which fault you are referring to in b). The vein in b) would not develop in that orientation if sigma 1 were horizontal. Also, keep in mind that the timing of cementation of the veins by postkinematic cements (see Ukar and Laubach, 2016) postdates timing of opening of the fracture. Don't mix the two! I had no idea that CV4 is restricted to the MdFF until now, because it is not mentioned anywhere in the text. How do you reconcile D3 to be surrounding breccia clasts within the MdFF in this model and sequence? Addressed

Estibalitz Ukar
Research Associate
Jackson School of Geosciences
The University of Texas at Austin

Dear Dr. Hendry and Dr. Ukar,

Thank you for the very constructive comments. They not only helped us to improve the quality of the manuscript but also our knowledge of dolomitizataion process. We have tried our best to address your suggestions in this new version of the manuscript.

Regarding the advancement of our research in comparison with the work previously presented by Ronchi et al. (2003), the current study gives much more details about the dolomite characterization and their relation to the structural evolution of the anticline on the regional and local scale. Furthermore, the obtained geochemical and microthermometry analyses do not confirm the role of marly or shaly basinal successions in providing the Mg-rich fluids during the first event of dolomitization (i.e. syn-rift), as proposed by Ronchi et al. (2003). We have tried to be modest in criticizing the latter authors limited research since the current research was build up on their findings. Another important question about the studied dolomites was the role of Scaglia Formation in providing the Mg-rich fluids during compression, because this formation is juxtaposed with the dolostones by the Montagna dei Fiori Fault. Our results do not support this hypothesis.

During our research, we also performed some other advance analyses such as clumped isotopes and U/Pb dating. However, the consecutive overgrowth pattern of dolomites and difficulties in isolating them to get enough and good quality samples increased the uncertainty in the results. Therefore, we decided not to include those data in the manuscript.

In the Montagna dei Fiori Anticline, the structures and their relative chronology are very complicated. A comprehensive structural study on the evolution of the Montagna dei Fiori Anticline was performed parallel with the current study, and published by Storti et al. (2018) in Tectonics. The target of the current study was to focus on dolomitization, and to use the structural model proposed by Storti et al. (2018) to deduce the most likely timing for dolomitization.

The distribution of dolomitization and sampled locations are way larger than the out crop photo scale. The dolomitized intervals are tens of meters mostly exposed in vertical to subvertical outcrops. To be able to show their 3D distribution properly a photogrammetry or LiDAR imaging is required.

The brecciated zones are mostly clast-support with minor calcite and negligible dolomite cement. Moreover, a detailed classification of breccia is not the focus of this research and does not give relevant information regarding this dolomitization case study.

This new version of the manuscript has been reviewed by a native English speaker.

Best regards,

Mozafari et al.

[revised manuscript text omitted]
 theabundant pressure solutions solution seems (TS), indicated by arrows, and their abutting relationship withcross-cutting calcite veins (CV2C2). Cc) A transmitted light photomicrograph of the dolomitized, brecciated Calcare Massiccio Formation. Note all the breccia fragments are composed of dolomite (D4 here).

Fig. 4. Field photographs (Corano Quarry) showing the field relations between dolostones (only D3 here), host limestones and the Montagna dei Fiori Fault: Aa) Panoramic view showing the spatial relationship between limestones and dolostones (orange) in the damage zone of the Montagna dei Fiori Fault (F). Note that the limestones and including dolostones of the Calcare Massiccio and Bugarone Formations on the footwall (FW) and marly limestones of the Scaglia Formation on the hangingwall (HW) are intensely deformed. Bb) Plan view of the dolomitized Calcare Massiccio limestone in the footwall damage zone: intersected by calcite veins (CV1C1), ) which are partially dolomitized, and affected by bed bed-perpendicular stylolites (arrows). Cc) Distinct transition (dashed line) between dolomitized and undolomitized Calcare Massiccio limestone in the footwall damage zone.

Fig. 5. Field photograph (Aa) and a simplified sketch (Bb) in field site d showing of a dolomitic pocket (grey color) within the folded Calcare Massiccio (grey color) and their its relation with bed bed-parallel stylolites within the Calcare Massiccio Formation (hammer is 40 cm long). Note C1 is the only calcite cement here.

[revised manuscript text omitted]

Fig. 10. Field photographs showing the major calcite vein settings observed in Montagna dei Fiori: Aa) Cross-sectional view of bed normal Calcite vein 1 (CV1C1) abutting bed bed-parallel stylolites in folded beds of the Calcare Massiccio Formation. Bb) Plan view of the Calcite vein 2 (CV2C2) intensely affecting the deformed Scaglia (Rossa) Formation. Cc, Dd) Cross-sectional view of the Scaglia Formation, intensely affected by pressure solution seams of tectonic origin crossed-over by populations of bed-perpendicular Calcite veins (CV3C3) in en echelon extensional arrays.

Fig. 11. Aa) Cathodoluminescence and transmitted light (in set) image showing blocky to elongated crystals of CV1C1 with zoned CL pattern in the Corano Quarry site. Bb) Transmitted light image showing intensely twinned CV1C1 crystals overprinted by euhedral to subhedral crystals of D3 in the Corano Quarry site. Photomicrographs of respectively, transmitted light and corresponding cathodoluminescence image: Cc, Dd) CV2C2 in the Scaglia Formation abutted by a bed bed-perpendicular stylolite (indicated by white arrows and dashed line) in the Corano Quarry site. The crystals display blocky to fibrous morphologies, deformation twinning, and a similar orange luminescence pattern comparable withsimilar to the adjacent host rock. Ee, Ff)

[revised manuscript text omitted]

Table. 1

[Figure]

Fig. 1

[Figure]

**(a)**

385000  386000  (a)

ⓒ

Castel
Manfrino

ⓓ

ⓔ

ⓕ

Salinello
creek

ⓖ

Osso
Caprino a ⊦  b ⊦

ⓗ

N

m

Montagna dei Fiori
Fault

4734000

4733000

4732000

**(b)**

a  1500  1500  b

1000

500

*Bisciaro
and Marne con Cerrogna Fms.*
*Scaglia Fm.*
*Marne a Fucoidi Fm.*
*Maiolica Fm.*
*Calcari ad Aptici*
*Salinello*
*Rosso Ammonitico*
*a  b  Corniola Fm.
limestone (a) and dolostone (b)*
*a  b  Calcare Massiccio Fm.
limestone (a) and dolostone (b)*

Fig. 2

[Figure]

(a)

Scaglia Fm.
Fig. 3b

Dolomitized
Calcare Massiccio Fm.
Fig. 3c

HW

FW

Montagna dei Fiori
Fault plane (b)

TS

(c)

μm

Fig. 3

[Figure]

Fig. 4

[Figure]

Fig. 5

[Figure]

Fig. 6

[Figure]

Fig. 7

[Figure]

Fig. 8

[Figure]

Fig. 9

[Figure]

Fig. 10

[Figure]

Fig. 11

[Figure]

Fig. 12

[Figure]

Fig. 13

[Figure]

Fig. 14

[Figure]

(a)

Normal faulting

Sea level

MAS

MAS

COI

BUN

Basement

BUN-Burano evaporites    MAS-Calcare Massiccio    COI-Corniola

D2

D1

BS

Syn-rift stage (b)

Hinterland

Trajectory of the
future fault

BUN

Basement

D3

SD

TS

D4

Layer parallel shortening (pre-early folding/ faulting)

(c)

Montagna dei Fiori Fault

D5

BUN    BUN

Backlimb collapse (late stage extensional faulting)

Fig. 15

**Review of "Fault-related dolomitization in the Montagna dei Fiori Anticline (Central Apennines, Italy): Record of a dominantly pre-orogenic fluid migration. Se-2018-136**

This manuscript presents field, petrographic and geochemical data from non-stratabound dolomites in a complex tectonic setting, and interprets their geofluid origin (parent fluids, timing) in the context of the tectonostratigraphic history.

It is coherent and logically organised. Aspects of the written English need minor improvement (punctuation, plurals, word order, etc); it will benefit from a final revision by a native English speaker.

The data are generally of good quality, and the interpretations are mostly justified from the results presented. In any study such as this, with limitations imposed by the ability to sample all the phases, there is necessarily some latitude or flexibility in the deductions that can be made. However, the authors do a good job of considering alternative possibilities for the fluid sources and timings.

My only issue with the paper is that the authors have not really considered whether there are wider implications or generic advances that can be made from the research. It presents itself as a case study, albeit one with a good integration of structural and diagenetic data. But what is the wider impact that will attract a non-specialist readership? Within the paper the authors all but admit that their findings are only modestly advanced from those of Ronchi and co-workers fifteen years ago (lines 638-642). I had hoped to see more progression in the science, and maybe the authors need to more thoroughly and critically evaluate the Ronchi model in the light of their new data. They could also work the structural data more – rather than just considering fault orientations and timings, what about the character and extent of the damage zones associated with different fault types / generations, and their relationship to the size and shape of dolomite bodies? What determines the lateral extent of dolomites? Is it other faults / fractures, or a gradual reaction front?

One generic aspect that the authors could address is implications for reservoir potential in analogues for this setting. The preponderance of planar dolomite is significant because planar dolomite is usually very beneficial for porperm (unlike many examples of hydrothermal dolomitization that feature tight nonplanar dolomites). Are there dolomitised plays in the Middle East that this study could be compared to (Zagros Mountains for example?), or maybe in Mexico?

Another factor of interest, largely by-passed in the text, is what drove the fluid circulation necessary to cause massive dolomitization when the low temperatures argue against a hydrothermal syn-rift system. Can the authors attempt a mass balance to estimate the order of magnitude fluid volumes needed? Maybe the dolomitization occurred in the down-flowing (cool) limb of a convection cell established on syn-rift faults that breached contemporary sea bed? Or does the dolomite zoning imply a pulsed fluid flow associated with strain cycling or seismic valving? I recommend the recent papers by Hollis and others on the Hammam Faraun fault and related syn-rift dolomitization. Are the D1-2 dolomites formed in a similar manner to this geologically younger example? Likewise, with the later dolomitization, which structures would have been open during compressional tectonics and able to serve as conduits for substantial fluid volumes?

If the authors address these issues their paper, which is already technically good, it will have much greater impact and interest across the sedimentary and structural geoscience community.

I have some more **specific comments** – these are tagged by line number or Figure number:

Line 43: The paper describes the role of evaporite-sourced fluids in the dolomitization process, but I am not sure that it illustrates a controlling role of evaporitic detachments. These may have influenced the tectonic development, but if it is believed that they directly controlled the dolomitization this needs to be specifically discussed later in the paper. Emphasized more in the text

Note the abstract is quite long-winded. It would be good to make it more succinct and punchier. Addressed

Line 69: The Castel Manfrino Dolostones are not labelled in Fig. 1 or Fig. 2b. Added in Fig.1

Line 73, 76: Did Ronchi (2003) base her study on the mapping of Mattei (1987)? Maybe there needs to be a couple of sentences describing Ronchi's findings so that it can be more clearly shown that the understanding has moved on. Added to text

Line 79-83: This is very long-winded and vague. It either needs to be shortened or to include specific details. The sentence is shortened.

Line 129: Given that the early dolomitization (D1-2) is later ascribed to the syn-rift stage, it would be useful to briefly describe the facies and architectural character of the syn-rift carbonates. For example, were they preferentially developed on footwall highs, in which case there was likely a juxtaposition of permeable high energy facies against faults that later hosted fluid flow? Addressed

Line 135-137: It may be a matter of debate, but the authors need to either provide the conflicting evidence or at least express a view and justify it. The sentence is deleted as further discussion is not the focus of this research

Line 152: Can the Montagna dei Fiori fault be indicated / labelled on Fig. 1d? Addressed

Line 164: Ground truthing implies that the features had previously been mapped out using remote data. If so, this should be included in the methods. Addressed

Line 167: So far as I can see, the Sibley and Gregg (1987) terminology (planar-e, planar-s, nonplanar) is <not> used anywhere in the paper, so either it needs to be incorporated or this sentence should be removed. The sentence and related references are removed

Line 186: Can the reproducibility (±0.1‰) be smaller than the precision (±0.2‰)? The inter-lab reproducability is ±0.1‰  means the measurement for the same sample in two different labs has a difference of ±0.1‰

Line 224: Bugarone Formation is not shown on Fig. 1 or Fig 2. Addressed
Line 231: Is the wider distribution of dolomitized intervals related to the topography of the valley and the exposures? If not, what is the relationship? Addressed

Line 238: I suggest not using "overprinted", which implies the original fabric / lithology is lost. Why not just use "cross-cut"? (or even just "cut") Addressed

Line 258 and elsewhere: "Dull" is not a colour! Addressed

Line 261 onwards: There could be a bit more detail on the dolomite distribution and fabric with respect to host rock facies. Is it all texturally destructive, is there any textural or mineralogical selectivity, were grainy or muddy facies preferentially dolomitized (controls by permeability versus reactive surface area…….?) Addressed

Line 272: There is an issue because CV1, CV2 etc. are introduced before they have been defined and described. I suggest starting section 4.2 with a paragenetic summary to alleviate this problem. Addressed, paragenesis is presented in Fig. 14.

Line 278: By using "frequently" the text suggests that sometimes (infrequently) D2 post-dates bed-parallel stylolites. Is that the case? Addressed

Line 284-285: This sentence needs a figure citation. Addressed

Line 297: Repetition of "euhedral to anhedral" (cf. line 295) – note this is not Sibley and Gregg terminology. Nor is "tightly packed texture" in line 279. Addressed

Line 305: I do not think one dolomite can "recrystallize" another. Recrystallization is a solid-state process that increased mineral stability. To demonstrate it might need data on the ordering, crystallinity and stoichiometry of D1/2 versus D4 (do the authors have any XRD data?). What is more likely is that D4 has locally replaced D1 and D2 by a dissolution-precipitation mechanism. However, the text lacks a clear description of the evidence for replacement. I recall papers by Mazzullo and by Machel that discuss this – it would be good to list the criteria for this case. Addressed

Line 330: In Fig. 11C, D the dolomite does not appear to be yellow-orange, it looks more like orange-brown. Addressed

Line 332: How wide was the extensional fault master plane? Please supply the range of widths (and lengths where possible) of the different vein generations. It is addressed for vein generations.

Line 335-336: What is meant by "with no evidence of physical disruption"? Does it mean that CV3 always passively overgrows D5 in voids and never cuts it? If so, it is easier to say this. Addessed

Line 336: Translucent is not a colour. Addressed

Line 334: What colours are the zones? Addressed

Line 367-368: Rather than "the presumable Lower Jurassic marine dolomites" – which are hypothetical – it is better to say the values are lower than those expected for Lower Jurassic seawater dolomites. Addressed

Line 391: "While….." indicates there should be a second part to the sentence. Addressed

Line 396: What is the lithology of these samples? Addressed

Line 401: Please add a sentence or two on the fluid inclusion petrography and distribution – do inclusions follow growth zones or are they randomly distributed? Are they all primary or are some pseudosecondary? What are their shapes and what are the liquid:vapour ratios in the 2-phase examples? Also, in reporting the results for different cements please give the number of inclusions the ranges are based on (n=). Addressed

Line 409-410: What is the purpose of nucleating a bubble for measurement of freezing temperatures? Addressed

Line 477-478: This sentence appears out of place or at least needs clarification. More than two values are needed to demonstrate a progression. Addressed

Line 503-507: Yes, this makes sense if the veins are filling tension gashes associated with stylolites – such as system is likely to be buffered by the dissolving carbonate. Maybe make this point more explicitly, and contrast with vein types that were more extensive and would have allowed allochthonous fluids to pass through with minimal host-rock interaction. Addressed

Line 518: Hendry et al. (2015) did not discuss $^{13}C$ enrichment from $CO_2$ outgassing due to evaporation. They made the point that negative covariance in C and O isotopes within veins could be due to precipitation during $CO_2$ outgassing related to pressure changes. Addressed

Line 524-551: This is good but is a very long paragraph. Can it be made more succinct? Addressed

Line 550-551: The final sentence needs rewriting; what was confined, the thrust wedge or the fluids? Addressed

Line 556: In the preceding section there is very little mention of fluid mixing. Could the poorly correlated Th, salinity and stable isotopic data reflect precipitation from allochthonous fluids as they mixed with in situ fluids (and cooled)? Degrees of mixing (and of water-rock interaction) may have different from fault to fault – is it really likely that the hydrogeological systems was as simple as is being presented here? The obtained data show no systematic variations from fault to fault. Thus, existance of different local hydrological systems cannot be addressed. The sentence is revised ad completed

Line 575-579: Please rewrite this sentence – it tries to say too many things at the same time. Addressed

Line 584: Doesn't the displacement of D1-2 on these faults indicate that the dolomite formed before faulting? What is the critical evidence that it is genuinely syn-rift? The syn-rift deposits (Corniola Formation) is affected by these dolomites. Addressed more in the text

Line 590: if D1-2 were related to basement-cutting faults, why are the Sr isotope values much less elevated than for D3 and D4? There is no other alternative for radiogenic Sr source. This is the case for all of the dolomite types. Maybe less basement deriven fluids were involved in D1.

Line 656-657: Please explain how hydrothermal fluids were able to circulate in the compressional tectonic regime – which structures were able to be in tension and therefore transmissive rather than sealing? Addressed

Line 1139: The cross-cutting relationship in Fig. 8a, b is not very evident. Addressed

Lines 1142-1148: Should this discussion be in the main text rather than in the caption? This has been made to avoid a longer discussion.

Fig. 2b: It would help if the colours and ornaments matched Fig. 8a (e.g. Salinello Fm). The text size needs to be increased for better legibility. Addressed

The stereonet data are good, but very little use is made of them in the text of the paper. Addressed in the figure caption

Fig. 5: Please increase the text size and make 5c larger – it is too small to see clearly. Addressed

Figs 6-9, 11: Some of the CL images could be a bit sharper and maybe with increased contrast to better discriminate the dolomite types. Addressed

Fig. 12: The symbols for D3 vs. CV3 and mixed dolomite vs. CV1 are too similar (especially given the small size). I am also not clear how Fig. 12b relates to Fig. 12a; maybe split the legend between the two plots according to what is in them – that might help. Addressed

Fig. 14: How were the burial temperatures in the burial history determined? What assumptions are they based on? Addressed

Fig. 15: I like this figure but I'm still not sure what the fluid flow pathway is in (b). Maybe a broader tectonic context diagram is needed showing expulsion of fluids from the foreland (if that is where they are coming from?). The fluids were migrated from hinterland (now indicated on the sketch) rather than forland.

I hope these comments are helpful, and I look forward to seeing the paper published in revised form.

Jim Hendry

Tullow Oil Ltd, Dublin

**Specific comments:**

1) Abstract: The abstract can be shortened substantially, yet it is missing key information. It provides too much detail of some aspects of this work but lacks equivalent detail in other cases. For example, why are calcite-filled veins not mentioned here? Weren't they a main focus of this study? A more succinct and balanced abstract is required. Also, the abstract would benefit from a "punchline" or statement of the broader implications of this work at the end. What did you learn about the extent of dolomitization near faults? How is this relevant for porosity/permeability evolution and fluid flow in dolomitized, carbonate-hosted hydrocarbon reservoirs and aquifers? Addressed

2) Introduction: You may want to consider adding a short statement about why some fault-zones become dolomitized but others don't. What are the requirements? What can you learn from outcrops that you cannot from core alone? I would say that the main benefit would be the opportunity to assess the spatial distribution of dolomitized zones, and individual diagenetic events, in 3D. Addition of such a field-relations analysis would greatly improve the impact of this work. Addressed

3) Geologic setting: This section is a bit long and could be shortened. Addressed

4) Methodology: A few things are missing:
   o How large of a geographic area did you sample? Addressed
   o How did you decide which areas to sample for isotopic analyses? Did you image them first? How? How confident are you that you didn't mix different cements when sampling? Addressed
   o How many fluid inclusions in your FIAs? What was your reproducibility and error? How did you make sure you did not measure stretched inclusions? Addressed

   o Documentation of where hand samples came from is very poor. This can be improved by showing the location of the thin sections on outcrop photos, and their spatial relationship with faults etc. These might need to be included in an appendix due to space restrictions, but it is important. Explained in author response

5) Field observations
   o What is the spatial distribution of the dolomitized geobodies? Addressed
   o And of the veined sections?
   o 6 outcrop locations are marked on Figure 2 but distributions of the different types of cements are only shown in one image (figure 5b). These are very important relations to assess fluid pathways and the evolution of dolomitization. Addressed
   o What are the orientations of CV1–CV4 cement-bearing fractures? Shown in Storti et al. (2018)

6) Petrography
   o How much calcite cement is there in the breccias? What are the textures? Why are these not included in your diagenetic evolution analysis? How do cements in breccias relate to those in host rocks? Are cements in the host rocks affected by brecciation? Show examples. Addressed

- How did dolomitization affect porosity in both host rock and fault rocks? How does porosity compare between limestones and dolostones? Addressed
- Some of the petrographic relationships mentioned need to be backed by images (see my line-specific comments) Addressed
- The order in which dolomite cements and vein-calcite cements are mentioned needs to be improved.
- What is the relationship between MC and D1/D2? Where is this documented? Addressed
- What is the distribution of CV cements in the host rock (see Laubach, 2003)? This should be properly documented and reported. I don't think vein cement is an appropriate term for these calcite cements. Also, keep in mind that the occlusion of fracture porosity by postkinematic cements can significantly postdate the timing of the opening of the fracture (see Ukar and Laubach, 2016). In other words: the timing of fracturing and cementation are not the same. Keep that in mind in your descriptions. Addressed
- The observation of CV3 in breccias is quite interesting. Document and show images. What other cements are there in breccias? How did you establish the relative timing of these cements and others? Breccia cements should not be referred to as vein cements. Explained in author response
- What are the spatial distributions of the different cement types? Addressed

7) Geochemistry
- What are the isotopic characteristics of MC and fibrous cements? Addressed
- Interpretations, especially for Sr ratios, should be moved to the discussion.

8) Fluid inclusions
- Show images of the different types of fluid inclusions, especially the FIAs.
- The graphs used to summarize fluid-inclusion thermometric results are not appropriate because key information is lost. Same for salinity. Please replot the data so that the temperature range for each individual FIA is shown. Did you measure an equal amount of FIAs for each type of cement? Otherwise, frequency would not very meaningful in Fig. 13 because it would be sample and cement availability dependent.
- Why are CV temperatures not shown in these graphs? For a higher focus on dolomitization case study

9) Discussion
- This section can be significantly shortened by avoiding repetition of results.
- I think parental fluid calculations should be shown in the results section, not in the discussion. To emphasize on the nature of parental fluids, we prefer to keep SMOW values in discussion section

- Use parallel writing style for stable isotopes and Sr discussion.???
- D3 shows significantly higher Sr/Sr than D4. How can both be related to the same event and derived from similar fault-related fluids? D1 and D2 are also fault-related. Why the differences in isotopic signatures, especially if all are related to basement-rooted faults??? Addressed

- The association of D3 and D4 with bed-parallel and shear fractures is mentioned for the first time in the discussion. This needs to be mentioned in the results. Are the cements themselves sheared? Show evidence. Addressed in petrography

- Discuss the spatial distribution of the different cement and vein/breccia types. What do they indicate about fluid-flow patterns? The breccia types in MDF are not diverse. More details on bereccia is out of focus of this study.

- The orientation of CV1–CV4-bearing fractures needs to be taken into account in the structural interpretation. A detailed structural interpretation is discussed in Storti et al. 2018. Its now more emphasized in this manuscript.

- Section 5.3: Without a better documentation of the orientations and field relations of the different cements and fracture types it is difficult to assess the validity of the inferred paragenetic sequence and the association of the different cements and structures with tectonic events. Some of the spatial and cross-cutting relationships between different types of cements are first mentioned in this section. Such descriptions should be moved to the results section.

- What is the driver for fluid circulation? Why are they Mg-rich fluids? What do fluid-inclusion salinities indicate? Addressed in section 5.2

- Why do you go from dolomite replacement and cementation to calcit cementation? Addressed

- How are your findings relevant for porosity/permeability evolution and fluid flow in dolomitized, carbonate-hosted hydrocarbon reservoirs and aquifers associated with similar reservoir-scale structures? Addressed

- How are your conclusions applicable to dolomitization processes associated with faults in general? How far can dolomitizing fluids travel and to what extent do they alter the mechanics and porosity/permeability of the host rock? What are the consequences for fluid-flow in these rocks? Addressed

10) Conclusions

- More thought needs to go into the conclusions section.

- I don't think enough data are presented in this study, especially of cross-cutting relationships and orientations of the different "deformation structures" to support the structural interpretation presented in the conclusions. For example, where is the evidence that the opening-mode fractures (no orientations or relationships within the anticline are reported!) and normal faults mentioned in this study are associated with contractional tectonics of the Apenninic orogeny? All are discussed in details in Storti et al. 2018.

11) Figures and figure captions need work, especially the model shown in Figure 15 (see comments in figure caption).

Lines 29-31: This needs to be re-written. Layer-parallel shortening would not give place to layer-parallel stylolites. Extensional faulting by itself either. Involvement in the Apenninic thrust wedge of what? Addressed

Lines 48-54: May I suggest you take a look at the recently published Ferraro et al. (2019) paper for a description of the diagenetic evolution of carbonate fault rocks in the central and southern Apennines? Addressed

Lines 54-58: You may want to consider adding a short statement about why some fault zones become dolomitized but others don't. What are the requirements? Addressed

Lines 58-61: What can you learn from outcrops that you cannot from core alone? I would say that the main benefit would be the opportunity to assess the spatial distribution of dolomitized zones, and individual diagenetic events, in 3D. Perhaps a missed opportunity in this work?

Line 67: I don't see Bugarone in Figure 1. Addressed

Line 69: I don't see catel Manfrino Dolostones in Figures 1 or 2. All the dolomitized intervals are called

Castel Manfrino Dolostones

Lines 72-73: How is your study better than Ronchi (2003)? Addressed

Lines 76-78: In Figure 2 it appears that dolomitized bodies are found quite far away from faults, beyond the typical lateral extent of fault damage zones. What is the explanation? Why are some faults associates with dolomitization while others aren't? Does it have to do with age of faults? Other factors? This would be a good topic for the discussion.

Line: 84: I would have liked to see more "field mapping" of the extent of D1–D5 and CV1–CV4 in this work.

Line 87: This sentence should start with a different word than "therefore". What provides insights? How? Addressed

Lines 88-89. Yes. This needs to be discussed in the discussion. Addressed

Lines 90-92: Yes. This needs to be discussed in the discussion. Addressed

Line 96: evolution of the Apennines has been proposed to be the result of Addressed

Line 98: since the Late Cretaceous Addressed

Line 103: The Central Apennines involve OK

Line 110: lower part of the Burano Formation Addressed

Line 115: Deposition of the Hettangian–Sinemurian Calcare Massiccio Formation, with a total thickness … Addressed

Line 117: following facies are present Addressed

Line 125: deepening-upward trend Addressed

Line 137: olistolith model This sentence is deleted

Lines 138-145: So, does this evidence favor the fault-related model or does this evidence provide an alternative model? Why does this sentence start with 'However"? This sentence is deleted

Line 151: at a high angle Addressed

Line 162-163: 60 samples distributed across how big of an area? Addressed

Line 164: What structures? Addressed

Line 187: Vienna Pee Dee Belemnite Addressed

Line 215: In order to perform high resolution Addressed

Lines 220-221: bed-perpendicular stylolites Addressed

Lines 224-230: This belongs in the Geological Setting. Addressed

Line 225: There is no evidence of dolomitization Addressed

Lines 231-234: Location names need to be included in figure captions. Addressed

Lines 235-239: This seems out of place. Start by describing mesoscale relations and distributions of dolomitized geobodies. Then focus on hand-sample and petrographic details. Addressed

Line 235: in fault cores are typically  Addressed

Line 237: is "main slip surface" a better term? Addressed

Line 238: cut by rather than overprinted. Are dolomite-filled veins intra- or intergranular?

Line 239: calcite cement Addressed

Line 241: cross-cutting bedding surfaces Addressed

Line 242: from a few meters to hundreds of meters Addressed

Line 243: and the lower part of  Addressed

Line 246: High amplitude (>1 mm), bed-parallel stylolites Addressed

Line 247-248: How does porosity differ between limestones and dolostones?

Line 253: grain-supported intervals Addressed

Lines 259-260: Evidence? Addressed

Line 265: we can't see the displacement mentioned in Figure 2A, site 1

Line 267-268: Are they overprinting or overgrowing? Show evidence. We also cannot see the distribution of the different cements at outcrop scale.

271-272: On what basis did you establish that the replacive dolomite within the host rock (D1) and lining fractures is the same?

Lines 275-276: solid inclusions of what? Insert figure call out for concentric zonation. Addressed

Line 281: sweeping extinction Addressed

Line 282: In some crystals, one… what types of solid and fluid inclusions? Addressed

Lines 286-291: Mark locations on map/figure captions and call out the figure. Addressed

Line 290-291: Scaglia Formation in the hanging wall. Addressed

Lines 293-295: This needs to be moved up Addressed

Line 305: bed-parallel shear fractures Addressed

Lines 308 on: There is a problem with CV introduction. What does it stand for? Must introduce them in chronological order. If the calcite cement is in veins it is most likely in the host rock as well (see Laubach, 2003). I don't think calling it vein cement is appropriate. Addressed

Line 308: What porosity? Addressed

Lines 318-320: Show outcrop photo?

Line 319: bed-perpendicular rather than bedding because that's what you use elsewhere. Make sure term usage is consistent throughout. Addressed

Line 321: bed-parallel stylolites. CV1 usually shows (often means time. Correct elsewhere in the manuscript). Addressed

Lines 324-326: Show image of CV2 in tension gashes Addressed

Line 332: extensional fault's master (main?) plane Addressed

Line 339-340: More evidence that the use of CV to refer to calcite cements that occur in a variety of textures and petrographic relations is not appropriate.

Lines 361-364: Why is this mentioned here and not with the rest of the calcite cements? Its ordered based on relative timig

Line 396: I see 3 values plotted for Scaglia. Addressed

Lines 398-399: Interpretation. Move to discussion. Same comment for previous paragraphs.Addressed

Lines 401-411: This belongs in the Methods section. Addressed

Lines 409 and 439: all-liquid inclusions Addressed

Lines 439-445: This belongs in the Methods section. Addressed

Lines 469-470: This belongs in the methods/results. Why did you avoid them? Could mottled D be a different type than those reported? Addressed

Lines 471-474: Move to methods. Report parental fluid calculations in the results section.

Line 478: Progressively higher than what? Fixed

Line 481: siliciclastic rocks,… Correct here and elsewhere. Addressed

Line 493: , or values recorded… Add references. Addressed

Lines 499 and 505: Replace comparable with similar. Here and elsewhere. Addressed

Line 507: stylolitization of the host rock (otherwise we do not know what dissolution etc. you are referring to). Addressed

Lines 527-528: fluids related to Late Messinian… overlying Upper Miocene Laga Formation and their possible…Addressed

Line 543: burial-related temperature Addressed

Line 544: it is unlikely that the.. Addressed

Line 546: located at higher stratigraphic levels Addressed

Line 564: calcite cements (FC) in grain-supported stratigraphic levels of the CMF is interpreted to be… Addressed

Line 570: bed-parallel stylolties Addressed

Line 574 and 575: are cut by Addressed

Line 579: Figure 15A call out. Addressed

Line 585: We cannot see the distribution described in Figure 2A.

Line 587: attributed to post-rift Addressed

Line 588: Although an absolute age cannot be provided, Addressed

Line 603: bed-parallel fractures. This is the first mention of shear veins for D4. Are the cements sheared? Show evidence. Cements are not sheared. Addressed

Line 604: Contractional deformations? How? Describe relationships better. Bed-perpendicular dilation alone would not cause shear.

Lines 608-610: First mention of this. Move to results.

Lines 618-619: bed-parallel veins Addressed

Line 622: fragments suggests that… late-stage evolution Addressed

Line 624: bed-perpendicular stylolites Addressed

Line 624-629. This is way too long. In any case, there is new information here that needs to be moved to the results section. Addressed

Line 629: low homogenization temperatures of fluid inclusions trapped within these cements Addressed

Lines 638-642: So, how is your study better? How are your conclusions applicable to dolomitization processes associated with faults in general?

Figures: Documentation of where samples came from is poor. Locations of samples need to be shown on outcrop photos and/or detailed maps. Add in appendix if space is limited.

Figure 1: Tiny name in a) is unreadable. Mark location of cross-section (A-A') shown in d).

Figure 2: Add names of each field site to the figure caption. Addressed

Figure 3: The picture in a) is too close up to see the context. Mark distribution of D1 etc. as in Fig 5d. Why isn't there more on these breccias (c) in the manuscript? Explain what arrows point to. Pressure solution seams. You are not showing intensity (it would be a number). Perhaps say showing abundant pressure solution seams. What are the abutting relationships? Which abuts which? Move arrow in b) so that the vein is actually visible. Are the other white pods also considered "veins"?

Figure 4: Show spatial distribution of D1, D3, CV1… etc. at the outcrop scale. The sentence seems to say that CV1 veins are dolomitized. Is that what you really mean? What is the dolomite type in b) and c)? Good opportunity to show cross-cutting relationships summarized in Figure 14.

Figure 5: What field site(s) and formation(s) are these from? In b) the zone shown in c) is marked as only having D1 but as D1 + D2 in c). Which one is correct? Any CV2-CV4 here? Addressed

Figure 6: What field site is this? rimmed by fibrous cements (FC), which are overgrown by mosaic cements (MC). overprinted. bed-parallel stylolites. D1cements lining a fracture. What do arrows point to in f) If it is D1, then what is to the right of it? Line 1123: which is cemented by CV1 in the center. Addressed

Figure 7: What field site(s) and formation(s) are these from? What is beyond D3 in e) and f)? Is D3 only present in breccias in this site? What is the CL signal of D3 in breccias and how did you establish correspondence with D3 in host rocks? What is the context of the sample? in e) and f) Addressed

Figure 8: What field site(s) and formation(s)? D3 and D4 are not cross-cutting but it appears that D4 overgrows D3. What is the D4 arrow in b) exactly pointing to? What cement is in the rhomb on the upper right corner? d) I am having a hard time seeing the microfracture. What CV is this? Or do you have dolomite-filled veins as well? Where are these described? What other cements are in e) and f) and what do arrows point to? Lines 1141-1150 belongs in the discussion. Addressed

Figure 9: What field site(s) and formation(s)? If D4 also occurs in fractures why is it not called vein cement as in your CV scheme? What other cements and/or host rock are in these photographs? Add labels. Addressed

Figure 10: These photographs are too close up to see the context. Outlining obscures fractures in a). c-d) These do not look like tension gashes to me (as mentioned in text?). Mislabeled as CV2 on the picture (?) but CV3 in the figure caption. Addressed

Figure 11: What field site(s) and formation(s)? What is a) a sample of? And the rest? Show field context. Addressed

Figure 12: Why aren't these figures in color? The symbols are too similar and they are hard to distinguish from one another. I would assign a color to each formation and a symbol to each diagenetic feature. Addressed

Figure 13: These plots are not very useful because key information is lost. Plot homogeneization and ice melting T ranges so that variability within individual FIAs is captured. Color would help, Also, where are the data for CV cements?

Figure 15: The fracture in a) would not have that orientation if sigma 1 were vertical. Indicate which fault you are referring to in b). The vein in b) would not develop in that orientation if sigma 1 were horizontal. Also, keep in mind that the timing of cementation of the veins by postkinematic cements (see Ukar and Laubach, 2016) postdates timing of opening of the fracture. Don't mix the two! I had no idea that CV4 is restricted to the MdFF until now, because it is not mentioned anywhere in the text. How do you reconcile D3 to be surrounding breccia clasts within the MdFF in this model and sequence? Addressed

Estibalitz Ukar
Research Associate
Jackson School of Geosciences
The University of Texas at Austin

Dear Dr. Hendry and Dr. Ukar,

Thank you for the very constructive comments. They not only helped us to improve the quality of the manuscript but also our knowledge of dolomitizataion process. We have tried our best to address your suggestions in this new version of the manuscript.

Regarding the advancement of our research in comparison with the work previously presented by Ronchi et al. (2003), the current study gives much more details about the dolomite characterization and their relation to the structural evolution of the anticline on the regional and local scale. Furthermore, the obtained geochemical and microthermometry analyses do not confirm the role of marly or shaly basinal successions in providing the Mg-rich fluids during the first event of dolomitization (i.e. syn-rift), as proposed by Ronchi et al. (2003). We have tried to be modest in criticizing the latter authors limited research since the current research was build up on their findings. Another important question about the studied dolomites was the role of Scaglia Formation in providing the Mg-rich fluids during compression, because this formation is juxtaposed with the dolostones by the Montagna dei Fiori Fault. Our results do not support this hypothesis.

During our research, we also performed some other advance analyses such as clumped isotopes and U/Pb dating. However, the consecutive overgrowth pattern of dolomites and difficulties in isolating them to get enough and good quality samples increased the uncertainty in the results. Therefore, we decided not to include those data in the manuscript.

In the Montagna dei Fiori Anticline, the structures and their relative chronology are very complicated. A comprehensive structural study on the evolution of the Montagna dei Fiori Anticline was performed parallel with the current study, and published by Storti et al. (2018) in Tectonics. The target of the current study was to focus on dolomitization, and to use the structural model proposed by Storti et al. (2018) to deduce the most likely timing for dolomitization.

The distribution of dolomitization and sampled locations are way larger than the out crop photo scale. The dolomitized intervals are tens of meters mostly exposed in vertical to subvertical outcrops. To be able to show their 3D distribution properly a photogrammetry or LiDAR imaging is required.

The brecciated zones are mostly clast-support with minor calcite and negligible dolomite cement. Moreover, a detailed classification of breccia is not the focus of this research and does not give relevant information regarding this dolomitization case study.

This new version of the manuscript has been reviewed by a native English speaker.

Best regards,

Mozafari et al.

[revised manuscript text omitted]
 theabundant pressure solutions solution seems (TS), indicated by arrows, and their abutting relationship withcross-cutting calcite veins (CV2C2). Cc) A transmitted light photomicrograph of the dolomitized, brecciated Calcare Massiccio Formation. Note all the breccia fragments are composed of dolomite (D4 here).

Fig. 4. Field photographs (Corano Quarry) showing the field relations between dolostones (only D3 here), host limestones and the Montagna dei Fiori Fault: Aa) Panoramic view showing the spatial relationship between limestones and dolostones (orange) in the damage zone of the Montagna dei Fiori Fault (F). Note that the limestones and including dolostones of the Calcare Massiccio and Bugarone Formations on the footwall (FW) and marly limestones of the Scaglia Formation on the hangingwall (HW) are intensely deformed. Bb) Plan view of the dolomitized Calcare Massiccio limestone in the footwall damage zone: intersected by calcite veins (CV1C1), ) which are partially dolomitized, and affected by bed bed-perpendicular stylolites (arrows). Cc) Distinct transition (dashed line) between dolomitized and undolomitized Calcare Massiccio limestone in the footwall damage zone.

Fig. 5. Field photograph (Aa) and a simplified sketch (Bb) in field site d showing of a dolomitic pocket (grey color) 
[revised manuscript text omitted]

Castel Manfrino

Salinello creek

Osso Caprino

Montagna dei Fiori Fault m

N

(a)

(b)

Fig. 2

[Figure]

Fig. 3

[Figure]

Fig. 4

[Figure]

Fig. 5

[Figure]

Fig. 6

[Figure]

Fig. 7

[Figure]

Fig. 8

[Figure]

Fig. 9

[Figure]

Fig. 10

[Figure]

Fig. 11

[Figure]

Fig. 12

[Figure]

Fig. 13

[Figure]

Fig. 14

[Figure]

Fig. 15